# Botulinum neurotoxin accurately separates tonic vs. phasic transmission and reveals heterosynaptic plasticity rules in *Drosophila*

Yifu Han[1], Chun Chien[1], Pragya Goel[1], Kaikai He[1], Cristian Pinales[2], Christopher Buser[2], Dion Dickman[1]*

[1]Department of Neurobiology, University of Southern California, Los Angeles, United States; [2]Oak Crest Institute of Science, Monrovia, United States

**Abstract** In developing and mature nervous systems, diverse neuronal subtypes innervate common targets to establish, maintain, and modify neural circuit function. A major challenge towards understanding the structural and functional architecture of neural circuits is to separate these inputs and determine their intrinsic and heterosynaptic relationships. The *Drosophila* larval neuromuscular junction is a powerful model system to study these questions, where two glutamatergic motor neurons, the strong phasic-like Is and weak tonic-like Ib, co-innervate individual muscle targets to coordinate locomotor behavior. However, complete neurotransmission from each input has never been electrophysiologically separated. We have employed a botulinum neurotoxin, BoNT-C, that eliminates both spontaneous and evoked neurotransmission without perturbing synaptic growth or structure, enabling the first approach that accurately isolates input-specific neurotransmission. Selective expression of BoNT-C in Is or Ib motor neurons disambiguates the functional properties of each input. Importantly, the blended values of Is+Ib neurotransmission can be fully recapitulated by isolated physiology from each input. Finally, selective silencing by BoNT-C does not induce heterosynaptic structural or functional plasticity at the convergent input. Thus, BoNT-C establishes the first approach to accurately separate neurotransmission between tonic vs. phasic neurons and defines heterosynaptic plasticity rules in a powerful model glutamatergic circuit.

*For correspondence:
dickman@usc.edu

## Editor's evaluation

This article reports a new genetically encoded neuronal silencer BoNT-C for use in *Drosophila*. The authors show that it fully blocks neurotransmission in two classes of *Drosophila* motor neurons (Is and Is; tonic and phasic, respectively). They also update a GCaMP postsynaptic reporter SynapG-CaMP8f. They selectively silence Ib or Is neurons to disambiguate the neurotransmission properties of each neuron, and finally, show that silencing either Ib or Is neurons does not induce heterosynaptic structural or functional plasticity. The data are convincing and the new silencing tool will be widely used.

## Introduction

Neural circuits are established in development, modified by experience, and must be maintained at stable physiological levels throughout the lifetime of an organism. Most cells embedded in neural circuits are innervated by multiple neurons, which differ in the number and strength of synaptic connections, the classes of neurotransmitters and neuropeptides released, and their patterns of activity.

There is also evidence that changes in synapse number or function at one input can induce adaptive or Hebbian modulations in transmission at convergent inputs, termed heterosynaptic plasticity (*Aponte-Santiago et al., 2020*; *Chater and Goda, 2021*; *Dittman and Regehr, 1997*; *Wang et al., 2021*). The *Drosophila* larval neuromuscular junction (NMJ) is a powerful model system for revealing fundamental principles of transmission and heterosynaptic plasticity, where synaptic development, growth, function, and plasticity have been studied for over 40 years (*Brunner and O'Kane, 1997*; *Charng et al., 2014*; *Harris and Littleton, 2015*). Most muscles at this glutamatergic synapse are co-innervated by two distinct motor neurons (MNs) that coordinate muscle contraction to drive locomotor behavior, termed MN-Is and MN-Ib. These MNs differ in both structural and functional properties, with the strong MN-Is firing with phasic-like patterns and depressing with repeated stimulation, while the weak MN-Ib fires with tonic-like patterns and facilitates (*Aponte-Santiago and Littleton, 2020*; *Hoang and Chiba, 2001*; *Lnenicka and Keshishian, 2000*; *Lu et al., 2016*; *Newman et al., 2017*). The vast majority of fly NMJ studies of synaptic structure have focused on MN-Ib terminals, given their larger relative size and amenability to light and super-resolution imaging (*Lu and Lichtman, 2007*; *Maglione and Sigrist, 2013*; *Sigrist and Sabatini, 2012*). Similarly, most studies using electrophysiology in this system have recorded an ambiguous blend of miniature events originating from both MN-Is and MN-Ib, as well as a composite evoked response from simultaneous stimulation of both inputs. It bears emphasizing that this compound-evoked response does not reflect the physiology of any actual existent synapse. This failure to fully disambiguate transmission from the strong MN-Is and weak MN-Ib has limited our understanding and ability to interpret synaptic transmission and plasticity at the *Drosophila* NMJ.

Although most electrophysiological studies using the *Drosophila* NMJ have failed to cleanly separate transmission from entire MN-Is and MN-Ib inputs, important insights have been gleaned into their respective properties. Early anatomical studies characterized their relative size and structures, which established that MN-Is boutons are smaller and contain fewer release sites, while those of MN-Ib are larger and opposed by a more elaborate subsynaptic reticulum (*Atwood et al., 1993*; *Jia et al., 1993*; *Lnenicka and Keshishian, 2000*; *Schuster et al., 1996*). Differences were also observed in the relative abundance of postsynaptic glutamate receptor subtypes at MN-Is vs. MN-Ib NMJs (*Marrus et al., 2004*; *Schmid et al., 2008*). Macro patch recordings at identified boutons suggested spontaneous events were larger at MN-Is terminals (*Pawlu et al., 2004*), consistent with electron microscopy (EM) studies that showed synaptic vesicles were larger at MN-Is terminals (*Karunanithi et al., 2002*). Finally, threshold stimulus manipulations established that single action potential-evoked transmitter release from MN-Is drives most of the depolarization in the postsynaptic muscle (*Lu et al., 2016*). Together, this important body of work detailed fundamental properties of the two MN subtypes in *Drosophila*.

Recent innovations in $Ca^{2+}$ imaging and the identification of GAL4 drivers that selectively express in MN-Is vs. MN-Ib have enabled new attempts to isolate transmission between the two inputs. First, postsynaptic $Ca^{2+}$ sensors were developed that revealed important differences in quantal release events, active zone-specific release characteristics, and plasticity between MN-Is vs. MN-Ib NMJs (*Akbergenova et al., 2018*; *Gratz et al., 2019Newman et al., 2022*; *Newman et al., 2017*; *Peled and Isacoff, 2011*). Following the recent identification of selective GAL4 drivers that target MN-Is and MN-Ib (*Aponte-Santiago et al., 2020*; *Pérez-Moreno and O'Kane, 2019*), input-specific manipulations suggested heterosynaptic structural plasticity could be induced between convergent Is vs. Ib inputs (*Aponte-Santiago et al., 2020*; *Wang et al., 2021*). However, to what extent heterosynaptic functional plasticity was expressed was unclear due to an inability to fully isolate transmission from either input. Finally, selective optogenetic stimulation of MN-Is or MN-Ib has been used to estimate evoked neurotransmission (*Genç and Davis, 2019*; *Sauvola et al., 2021*), but is unable to resolve input-specific miniature transmission and may suffer from confounds due to chronic channel rhodopsin expression in MNs. An accurate electrophysiological understanding of transmission from entire MN-Is vs. MN-Ib NMJs has therefore remained unresolved due to the significant limitations of these approaches. Similarly, clear rules and mechanistic insights into the signaling systems mediating heterosynaptic plasticity at the fly NMJ have not been established.

We have therefore sought to develop an optimal approach to accurately separate neurotransmission between tonic and phasic motor inputs in *Drosophila*. Towards this goal, we screened a series of botulinum toxins for abilities to prevent neurotransmitter release from MNs. This approach identified botulinum neurotoxin-C (BoNT-C) to eliminate all spontaneous and evoked neurotransmitter

release without imposing apparent intrinsic toxicity or plasticity at the convergent input. BoNT-C now enables the accurate separation of motor inputs and establishes heterosynaptic plasticity rules at the *Drosophila* NMJ.

## Results

### Suboptimal approaches to disambiguate tonic and phasic inputs at the *Drosophila* NMJ

An ideal approach to functionally isolate MN-Is and MN-Ib would silence all neurotransmission from one input without inducing structural or functional plasticity from the convergent input ('heterosynaptic plasticity'). If such an approach were successful, then the frequency of miniature transmission should be reduced when either input is silenced compared to recordings from wild-type NMJs (where miniature transmission is blended), and quantal size should be increased at MN-Is NMJs relative to MN-Ib. In addition, synaptic strength, as assessed by single action potential stimulation, should be increased when evoked from MN-Is relative to MN-Ib NMJs. Importantly, when the composite values of miniature and evoked transmission from MN-Is and MN-Ib NMJs are averaged or summed, they should fully recapitulate all aspects of neurotransmission as assessed from standard wild-type NMJ recordings, where miniature events are a mix from both inputs and evoked transmission is an ambiguous average of simultaneous release from both MN-Is and MN-Ib.

The fly larval musculature is composed of 12 repeated segments, with ~30 muscles per abdominal hemisegment (60 per segment), innervated by 36 distinct MNs: ~30 MN-Ib and 3 MN-Is per hemisegment (; *Clark et al., 2018*; *Hoang and Chiba, 2001*; *Kim et al., 2009*; *Lnenicka and Keshishian, 2000*). MN-Ib inputs typically innervate single muscles, while MN-Is co-innervates groups of several muscles (*Figure 1—figure supplement 1*). To distinguish transmission between MN-Is and MN-Ib inputs at the *Drosophila* NMJ, we first confirmed expression profiles of four MN GAL4 drivers that express at the muscle 6/7 NMJ, the primary NMJ that has been used for electrophysiology in the field and that we will focus on in this study: (1) *OK6-GAL4*, which drives GAL4 expression in all MNs, including the ones that innervate muscle 6/7; (2) *OK319-GAL4*, which expresses in small subsets of both Is and Ib MNs, including the ones that innervate muscle 6/7; (3) 'Is-GAL4' (*R27E09-GAL4*), which expresses in the MN-Is that innervates muscle 6/7; and (4) 'Ib-GAL4' (*dHB9-GAL4*), which expresses in the single MN-Ib that innervates muscle 6/7. Expression of these drivers proved specific (*Figure 1—figure supplement 1*), where expression begins by at least early first-instar stages (*Figure 1—figure supplement 2*), motivating a genetic dissection of the MN-Ib and MN-Is inputs at the muscle 6/7 NMJ.

One promising approach to electrophysiologically separate MN-Is and MN-Ib would be to employ a conditional null allele of the *vesicular glutamate transporter* (*vGlut*) that was recently developed (*Banerjee et al., 2021*; *Sherer et al., 2020*). In principle, this would be an ideal approach because all glutamate release would be silenced without otherwise perturbing innervation or synaptogenesis. Indeed, conditional loss of *vGlut* eliminates all neurotransmission in adult NMJs (*Banerjee et al., 2021*). However, at larval NMJs we found *vGlut* expression to be only moderately reduced at presynaptic terminals and did not eliminate transmission, despite trying all four of the GAL4 drivers described above to conditionally remove *vGlut* (*Figure 1—figure supplement 3*). This is likely due to perdurance of maternally contributed *vGlut* and early *vGlut* synthesis, as well as the finding that a single vGlut transporter is sufficient to fill a synaptic vesicle (*Daniels et al., 2006*). Hence, conditional loss of *vGlut* in larval MNs was not an effective approach for our goal of eliminating miniature and evoked transmission.

Using the Is-GAL4 and Ib-GAL4 drivers described above, we tried three alternative approaches to distinguish transmission between these two inputs. Each of these was previously used in attempts to functionally separate MN-Is/MN-Ib transmission, and each proved inadequate for our purposes. First, we used selective optogenetic stimulation of MN-Ib or MN-Is as recently employed (*Genç and Davis, 2019*; *Sauvola et al., 2021*). From the outset, we knew the limitations of this approach in failing to distinguish input-specific miniature transmission and an inability to perform repeated trains of stimulation to assess short-term plasticity and resolve vesicle pools. We expressed *channel rhodopsin* (ChR) in all MNs (OK6>ChR) or selectively in either MN-Is or MN-Ib (*Figure 1A and B*), as performed in previous studies. As expected, miniature frequency and amplitude were indistinguishable in Is>ChR and Ib>ChR compared to OK6>ChR or wild-type (*Figure 1A and B*), while evoked

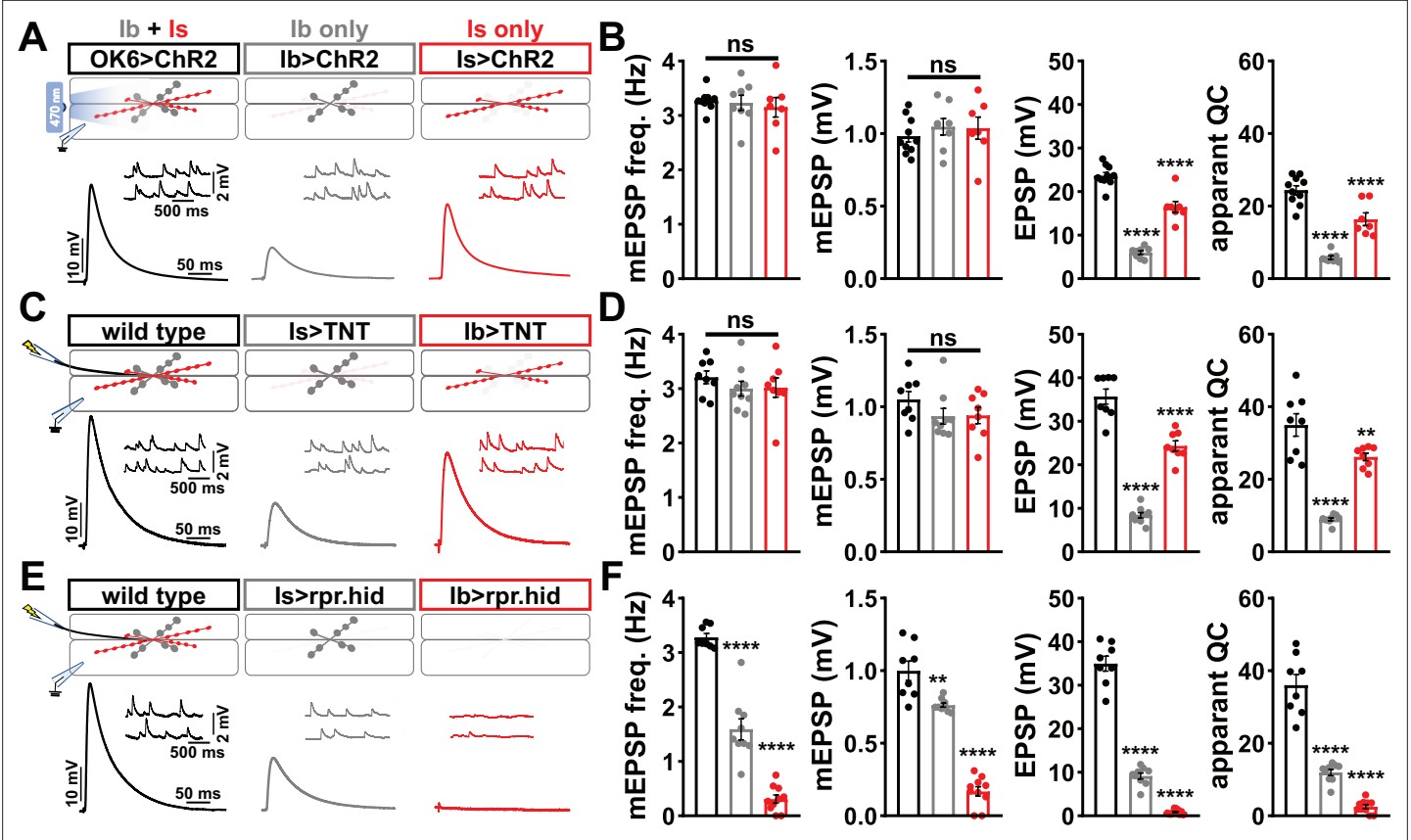

**Figure 1.** Suboptimal approaches for selectively isolating tonic and phasic neurotransmission at the *Drosophila* neuromuscular junction (NMJ). (**A**) Schematic of recording configuration and representative miniature excitatory postsynaptic potential (mEPSP) and excitatory postsynaptic potential (EPSP) traces illustrating that overexpression of *ChR2^T159C* in MN-Ib or -Is enables optically evoked EPSP events from either MN-Ib or -Is NMJs. Genotypes: OK6>ChR2 (*w;OK6-GAL4/UAS-ChR2^T159C;+*); Ib>ChR2 (*w;UAS- ChR2^T159C/+;dHB9-GAL4/+*); Is>ChR2 (*w;UAS- ChR2^T159C/+;R27E09-GAL4/+*). (**B**) Quantification of mEPSP frequency, amplitude, EPSP, and apparent quantal content values in the indicated genotypes in (**A**). Note that because input-specific mEPSP values cannot be determined using this optogenetic approach, inaccurate quantal content values are shown by simply dividing the EPSP values by the same averaged mEPSP values. (**C**) Schematic and representative traces following selective expression of tetanus toxin (TNT) in MN-Ib or -Is. Genotypes: wild-type (*w^1118*); Is>TNT (*w;+;R27E09-GAL4/UAS-TNT*); Ib>TNT (*w;+;dHB9-GAL4/UAS-TNT*). (**D**) Quantification of the indicated values of the genotypes shown in (**C**), with the same inaccuracies in determining quantal content. (**E**) Schematic and representative traces following input-specific expression of the pro-apoptotic genes *reaper* (*rpr*) and *head involution defective* (*hid*). Note that MN-Is NMJs are completely absent following MN-Ib ablation. Genotypes: wild-type (w); Is>rpr.hid (*UAS-rpr.hid/+;+; R27E09-GAL4/+*); Ib>rpr.hid (*UAS-rpr.hid/+;+; dHB9-GAL4/+*). (**F**) Quantification of the indicated values of the genotypes shown in (**E**). Error bars indicate ± SEM. ****p<0.0001; ***p<0.001; **p<0.01; ns, not significant. Additional statistical details are shown in ***Supplementary file 2***.

The online version of this article includes the following figure supplement(s) for figure 1:

**Figure supplement 1.** Motor neuron (MN)-specific GAL4 expression at the *Drosophila* neuromuscular junction (NMJ).

**Figure supplement 2.** MN-Is and MN-Ib drivers are expressed by early first-instar larval stages and block neurotransmission when crossed to botulinum neurotoxin (BoNT-C).

**Figure supplement 3.** vGlut expression persists despite conditional knockout in motor neurons.

**Figure supplement 4.** Electrophysiological differences between optogenetic and electrical stimulation at the *Drosophila* neuromuscular junction (NMJ).

**Figure supplement 5.** Genetic ablation of MN-Ib abolishes innervation by MN-Is on the same target.

release was stronger at Is compared to Ib, as expected. However, we also noted significant differences between optogenetic vs. electrical stimulation (***Figure 1B and D***, ***Figure 1—figure supplement 4***), where baseline-evoked excitatory postsynaptic potential (EPSP) values were reduced by ~33% in OK6>ChR compared to wild-type controls, suggesting that chronic expression of ChR alone induces significant changes in intrinsic excitability and synaptic function. Next, we expressed *tetanus toxin* [TNT; (***Sweeney et al., 1995***)] in an attempt to selectively silence evoked transmission in either MN (***Figure 1C and D***). Although TNT expression blocks evoked release, miniature transmission

persists (*Choi et al., 2014*; *Sweeney et al., 1995*), so we also knew from the outset that TNT, like optogenetic stimulation, would fail to separate miniature transmission. Indeed, selective expression of TNT does not significantly change miniature excitatory postsynaptic potential (mEPSP) frequency or amplitude from wild-type (*Figure 1C and D*). Further, while evoked EPSPs were larger at MN-Is NMJs compared to MN-Ib, as observed with optogenetic stimulation, input-specific TNT expression was reported to induce heterosynaptic plasticity (*Aponte-Santiago et al., 2020*), making it unclear if the input-specific differences in synaptic strength reflect genuine wild-type behavior. Finally, we tried genetic ablation of either MN-Is or MN-Ib by expression of the pro-apoptotic genes *reaper* (*rpr*) and *head involution defective* (*hid*). Is>rpr.hid cleanly eliminated MN-Is inputs (*Figure 1—figure supplement 5*), but was also reported to induce heterosynaptic plasticity (*Aponte-Santiago et al., 2020*; *Wang et al., 2021*). However, Ib>rpr.hid did not always completely ablate MN-Ib inputs, yet always fully eliminated all MN-Is inputs (*Figure 1—figure supplement 5*). While electrophysiological recordings from Is>rpr.hid appeared to silence transmission from MN-Is (*Figure 1E and F*), virtually all transmission was ablated at Ib>rpr.hid NMJs (*Figure 1E and F*). Thus, conditional mutagenesis, optogenetics, TNT expression, and genetic ablation each proved inadequate to accurately distinguish input-specific transmission.

## BoNT-C eliminates neurotransmission without confounding alterations in pre- or postsynaptic structure

We therefore sought to develop a new approach designed to eliminate all neurotransmission (both miniature and evoked release) and that ideally would not otherwise perturb MN structure or innervation. This search led us to botulinum neurotoxins (BoNTs), clostridial toxins that function as potent protein enzymes to cleave the synaptic vesicle SNARE complexes necessary for exocytosis (*Figure 2A and B*; *Dong et al., 2019*). We cloned sequences encoding four different BoNT light chains (BoNT-A, -B, -C, and -E), each targeting distinct SNARE components (*Figure 2A and B*) into *Drosophila* transgenic vectors under control of GAL4-responsive UAS sequences. Importantly, the cleavage targets of each BoNT were validated for the *Drosophila* proteins (*Backhaus et al., 2016*). From the outset, we suspected that BoNT-C would likely be an ideal candidate since it targets the t-SNARE Syntaxin, mutations of which in *Drosophila* eliminate all miniature and evoked release (*Deitcher et al., 1998*; *Schulze et al., 1995*). As a basis of comparison, miniature transmission persists in mutations targeting the *Drosophila* vSNARE *neuronal synaptobrevin* (*n-syb*; *Broadie et al., 1995*; *Deitcher et al., 1998*), and n-Syb is the target of TNT (n-Syb; *Figure 2A*). As an initial screen for SNARE cleavage activity, we evaluated lethality following pan-neuronal (*C155-GAL4*) and MN-specific (*OK6-GAL4*) BoNT expression. Following these filters, only BoNT-B and BoNT-C caused embryonic lethality (*Figure 2C*). To circumvent lethality, we then expressed either BoNT-B and BoNT-C with *OK319-GAL4*, where NMJ electrophysiological recordings revealed BoNT-B expression failed to fully eliminate transmission (*Figure 2C*, *Supplementary file 1*). However, while miniature transmission persists in OK319>TNT, OK319>BoNT-C eliminated both spontaneous and evoked neurotransmission (*Figure 2D and E*). Thus, BoNT-C expression in MNs was identified as the only SNARE toxin capable of eliminating both miniature and evoked neurotransmission.

Next, we examined whether BoNT-C expression perturbs presynaptic growth or structure when expressed in MNs. Neuronal toxicity has been reported for some classes of neurotoxins (*Peng et al., 2013*), and we continued to use TNT expression as a comparison. Expression of TNT in both MN-Is and -Ib by OK319>GAL4 led to an ~20% reduction in bouton and active zone number at terminals of MN-Ib (indicated by immunostaining of the active zone scaffold Bruchpilot [BRP]), while an inverse change in synaptic growth was observed in MN-Is (*Figure 3A*), consistent with previous reports (*Goel and Dickman, 2018*). In contrast, BoNT-C expression in both MN-Is and MN-Ib did not significantly change bouton or BRP puncta number, nor was BRP intensity altered in either input (*Figure 3B*). Similarly, we observed no differences in NMJ ultrastructure following BoNT-C expression, with no significant changes in T-bar or active zone length or synaptic vesicle density (*Figure 3C and D*). Although presynaptic homeostatic plasticity remodels active zone structure at the fly NMJ (*Böhme et al., 2019*; *Goel et al., 2019a*), induction of this form of plasticity does not require activity (*Frank et al., 2006*; *Goel et al., 2017*; *Goel and Dickman, 2021*; *Kikuma et al., 2019*; *Li et al., 2018*). Therefore, silencing of activity by BoNT-C would not be expected to change BRP structure. Thus, BoNT-C expression does not perturb presynaptic growth or structure.

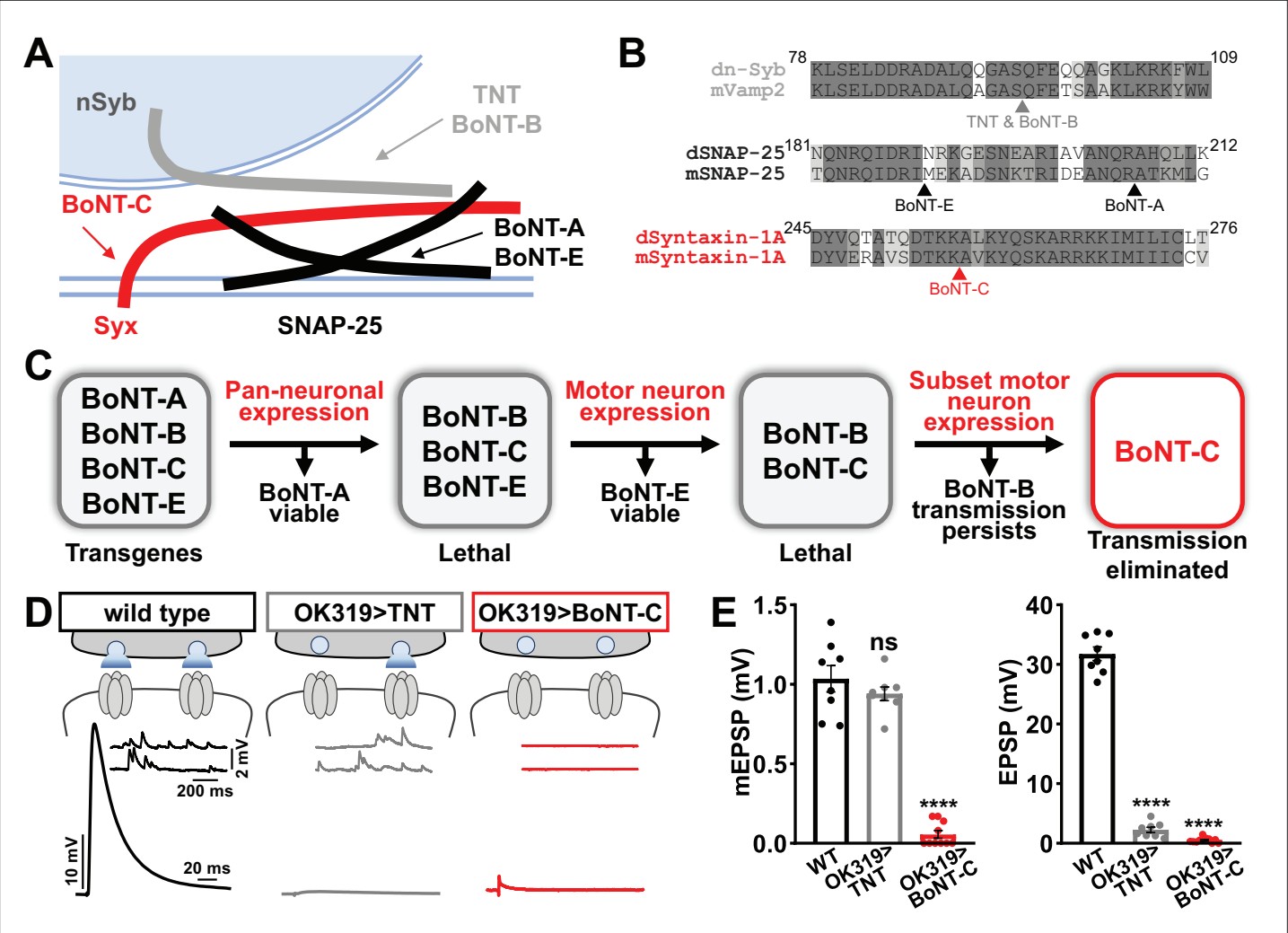

**Figure 2.** Botulinum neurotoxin (BoNT-C) eliminates both spontaneous and evoked transmission. (**A**) Schematic of synaptic SNARE proteins that are targets for enzymatic cleavage by the tetanus (TNT) and botulinum (BoNT) neurotoxins. (**B**) Amino acid sequence alignments of the cleavage sites for the indicated TNT and BoNT toxins in the mouse and *Drosophila* SNARE components synaptobrevin (Vamp2 and neuronal synaptobrevin [dN-Syb]), SNAP25, and Syntaxin (Syntaxin-1A). Arrows illustrate the conserved cleavage sites of TNT, BoNT-A, BoNT-B, BoNT-C, and BoNT-E. (**C**) BoNT screening flowchart: four BoNT lines were first tested for lethality when expressed using the pan-neuronal *c155-GAL4* driver, then similarly tested when crossed to the motor neuron-specific *OK6-GAL4* driver, resulting in only BoNT-B and BoNT-C causing lethality. These transgenes were then crossed to the *OK319-GAL4* driver, which expressed in only a subset of motor neurons, and electrophysiological recordings revealed that while transmission persisted in BoNT-B, transmission was completely blocked in BoNT-C. (**D**) Schematic and representative electrophysiological traces illustrating that while miniature transmission persists at neuromuscular junctions (NMJs) poisoned by TNT expression, transmission was completely blocked at NMJs expressing BoNT-C. (**E**) Quantification of miniature excitatory postsynaptic potential (mEPSP) and excitatory postsynaptic potential (EPSP) amplitudes in the indicated genotypes: wild-type (w); OK319>TNT (*w;OK319-GAL4/UAS-TNT;+*); OK319>BoNT-C (*w;OK319-GAL4/+;UAS- BoNT-C/+*). Error bars indicate ± SEM. ****$p<0.0001$; ns, not significant. Additional BoNT screening results and statistical details are shown in ***Supplementary file 1*** and ***Supplementary file 2***.

Finally, we determined whether BoNT-C expression in MNs altered postsynaptic structure at the NMJ. The postsynaptic glutamate receptors at the NMJ assemble as heterotetramers containing three essential subunits (GluRIII, GluRIID, and GluRIIE) and either a GluRIIA or GluRIIB subunit (***Han et al., 2015***; ***Qin et al., 2005***). GluRIIA-containing receptors drive most of the synaptic currents due to the rapid desensitization of GluRIIB-containing subtypes (***Diantonio et al., 1999***; ***Han et al., 2015***). Differences in GluR composition have been reported at postsynaptic compartments of MN-Is and MN-Ib, where GluRIIA-type receptors localize more centrally within GluR fields and are more abundant at MN-Ib postsynaptic compartments relative to MN-Is (***Akbergenova et al., 2018***; ***Marrus et al., 2004***; ***Schmid et al., 2008***). It has been speculated that tonic vs. phasic patterns of activity at MN-Is and MN-Ib may

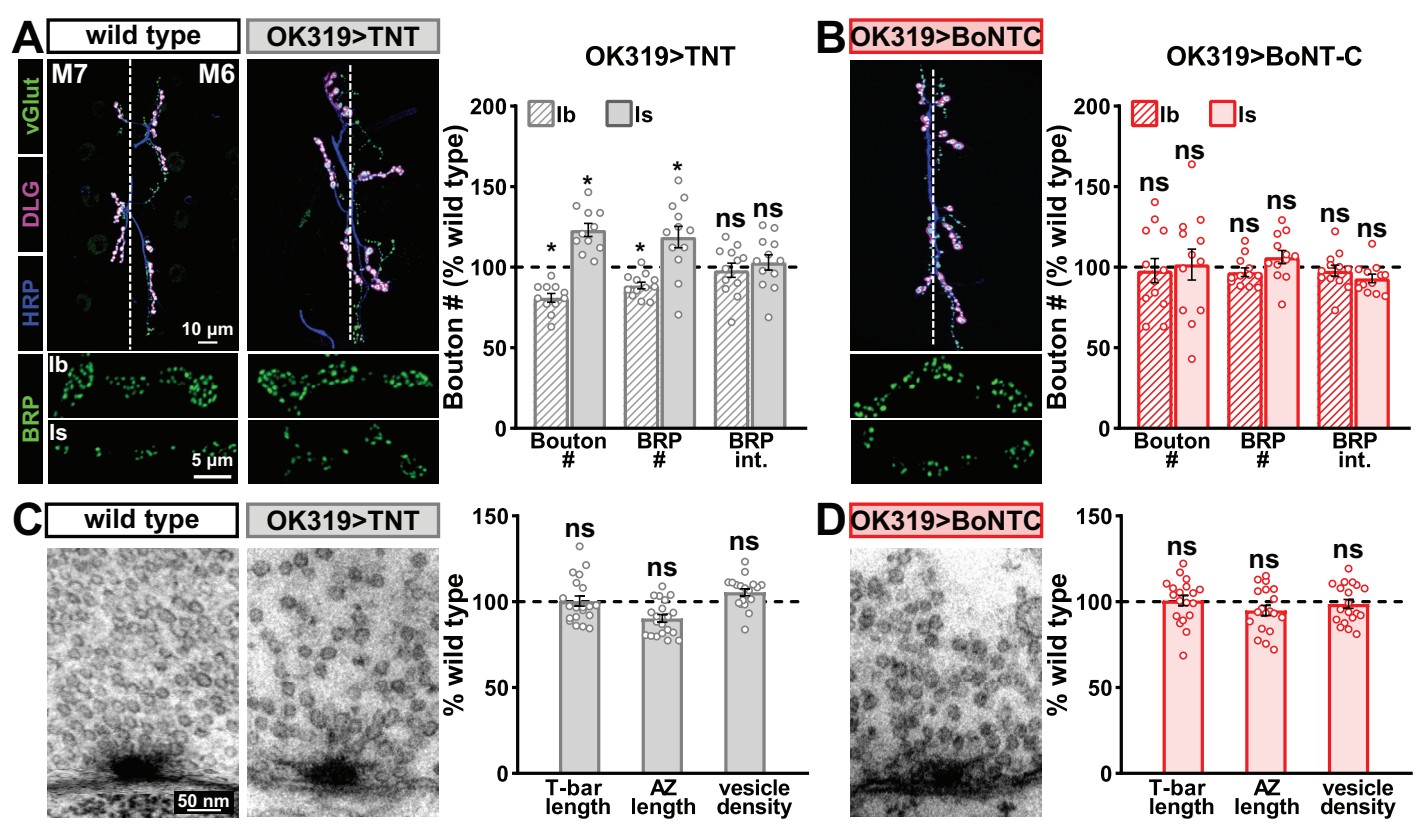

**Figure 3.** Botulinum neurotoxin (BoNT-C) expression does not perturb presynaptic growth or structure. (**A**) Representative muscle six neuromuscular junction (NMJ) images immunostained with anti-vGlut, -DLG, and -HRP at wild-type and NMJs expressing tetanus toxin (TNT); BRP immunostaining in individual boutons is shown below. Right: quantification of bouton number, BRP number/NMJ, and mean BRP fluorescence intensity at TNT NMJs of MN-Ib or -Is expressing normalized to wild-type values. (**B**) Representative images and quantification of NMJs silenced by BoNT-C expression as described in (**A**). Note that in contrast to TNT, BoNT-C expression does not change bouton or BRP numbers at NMJs of either MN-Ib or -Is. (**C**) Representative electron micrographs of wild-type and TNT NMJs showing synaptic vesicles and active zone structures. Right: quantification of T-bar length (μm), active zone length (μm), and synaptic vesicle density (#/μm²) normalized to wild-type values in the indicated genotypes. (**D**) Representative electron micrographs and analysis of BoNT-C NMJs as presented in (**C**). Note that no significant differences are observed compared to wild-type values. Error bars indicate ± SEM. *p<0.05; ns, not significant. Absolute values for normalized data and additional statistical details are summarized in ***Supplementary file 2***.

orchestrate these differences in GluR composition at postsynaptic receptive fields (***Aponte-Santiago and Littleton, 2020***). We used synaptic silencing by BoNT-C as an opportunity to test this possibility. At wild-type postsynaptic compartments, we observed the lower GluRIIA intensity characteristic at MN-Is NMJs relative to MN-Ib (***Figure 4A and B***). We also found the characteristic enrichment of GluRIIA at centers of receptive fields, with GluRIIB distributed in the periphery (***Figure 4C***). Interestingly, these same input-specific patterns of GluR composition and spatial localization were observed following BoNT-C expression, where no glutamate is emitted from presynaptic terminals (***Figure 4D–F***). We also did not find any major differences in GluR localization or maturation at earlier developmental stages following BoNT-C expression (***Figure 4—figure supplement 1***), although we cannot rule out the possibility of more subtle alterations in the timing of postsynaptic maturation. Nonetheless, these results demonstrate two important points. First, neurotransmitter release is not required for synaptic assembly and alignment at the *Drosophila* NMJ, nor for their maintenance during growth and elaboration, including the specialization of postsynaptic GluR fields. Second, patterns of tonic vs. phasic transmission do not sculpt the final composition of postsynaptic receptive fields.

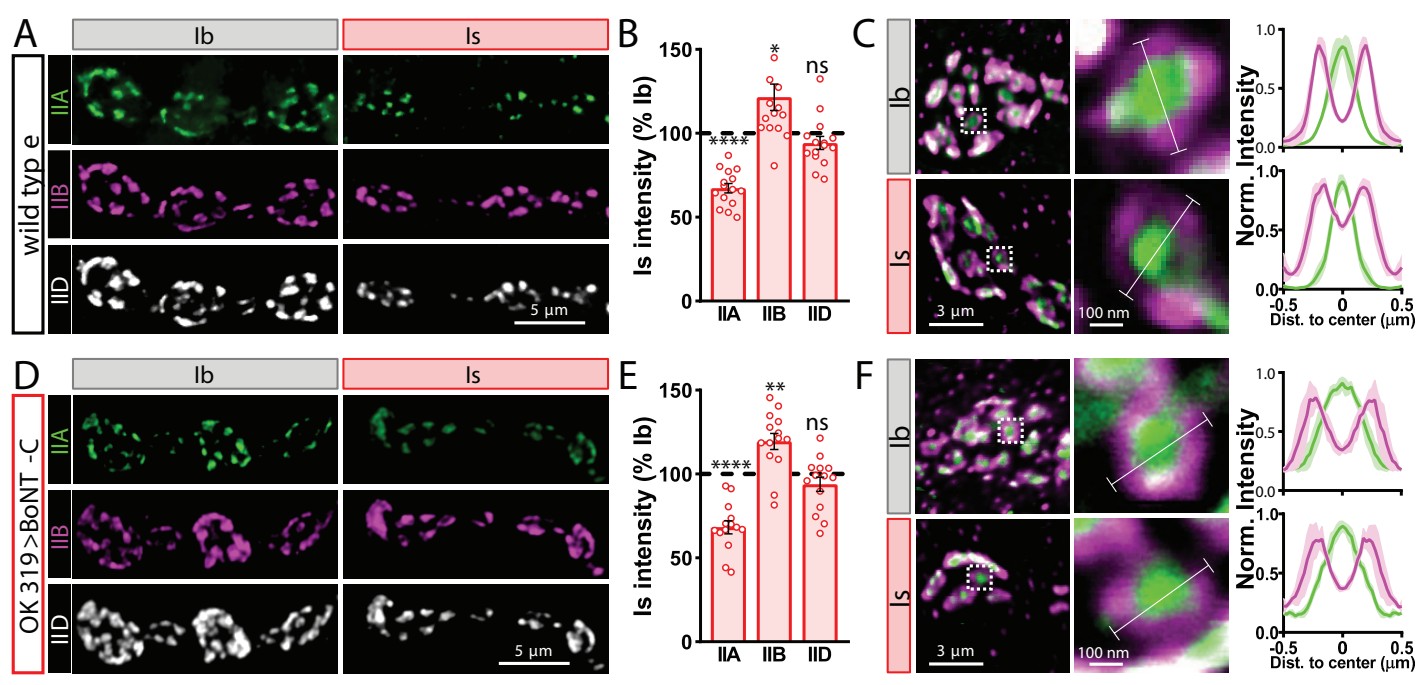

**Figure 4.** Tonic vs. phasic activity patterns do not specialize postsynaptic GluR fields. (**A**) Representative images of boutons from MN-Ib and -Is neuromuscular junctions (NMJs) immunostained with anti-GluRIIA, -GluRIIB, and -GluRIID in wild-type. (**B**) Quantification of the mean fluorescence intensity of each GluR subunit at MN-Is NMJs normalized as a percentage of those at MN-Ib. MN-Is NMJs exhibit a significant decrease in GluRIIA-containing GluRs, an increase in GluRIIB-containing GluRs, and no significant change in total GluR levels (indicated by the common subunit GluRIID) compared to GluR fields at MN-Ib NMJs. (**C**) High-magnification image of individual NMJs imaged as in (**A**). Averaged fluorescence line profiles show GluRIIA and GluRIIB normalized to peak fluorescence values across 10 receptor fields in MN-Ib or -Is NMJs. Note that peak fluorescence of GluRIIA is at the center of the GluR field, while peak fluorescence of GluRIIB is located more peripherally. The white line indicates the line profile region of interest (ROI). (**D–F**) Similar analysis of MN-Ib and -Is NMJs silenced by botulinum neurotoxin (BoNT-C) expression (OK319>BoNT-C). Similar GluR levels and localizations are observed, indicating that tonic vs. phasic patterns of transmission, and indeed glutamate release itself, are not required to establish the input-specific specialization of GluR fields. Error bars indicate ± SEM. ****p<0.0001; **p<0.01; *p<0.05; ns, not significant. Absolute values for normalized data and additional statistical details are summarized in *Supplementary file 2*.

The online version of this article includes the following figure supplement(s) for figure 4:

**Figure supplement 1.** Botulinum neurotoxin (BoNT-C) silencing does not delay postsynaptic maturation.

## Input-specific silencing by BoNT-C does not induce heterosynaptic structural plasticity

Next, we sought to determine whether selective silencing of neurotransmitter release at MN-Is or MN-Ib by BoNT-C induced heterosynaptic structural plasticity. Previous studies have found heterosynaptic changes in synaptic growth following genetic ablation or TNT expression (*Aponte-Santiago et al., 2020*; *Wang et al., 2021*). First, we engineered the newest and fastest genetically encoded Ca²⁺ sensor (GCaMP8f) into a previous generation postsynaptic sensor (SynapGCaMP6f; *Newman et al., 2017*) to visualize input-specific elimination of synaptic transmission the larval NMJ. When BoNT-C is expressed in MN-Is only, Ca²⁺ imaging using SynapGCaMP8f revealed the complete elimination of both miniature and evoked transmission at MN-Is NMJs, while spontaneous and evoked transmission was unperturbed at MN-Ib NMJs (*Figure 5A–C*). Conversely, selective BoNT-C expression in MN-Ib eliminated transmission from MN-Ib NMJs without impacting transmission from MN-Is (*Figure 5D–F*). Using the SynapGCaMP8f reporter, we also verified that BoNT-C expression blocked all synaptic activity in earlier developmental stages (*Figure 1—figure supplement 2*). Thus, Ca²⁺ imaging using SynapGCaMP8f confirmed that selective BoNT-C expression can induce input-specific synaptic silencing throughout early developmental stages.

We then examined synaptic growth by counting Is vs. Ib bouton numbers following input-specific expression of BoNT-C, TNT, and rpr.hid. When BoNT-C is expressed in MN-Is, no changes in synaptic growth in either MN-Is or MN-Ib were observed compared to wild-type (*Figure 6A and B*),

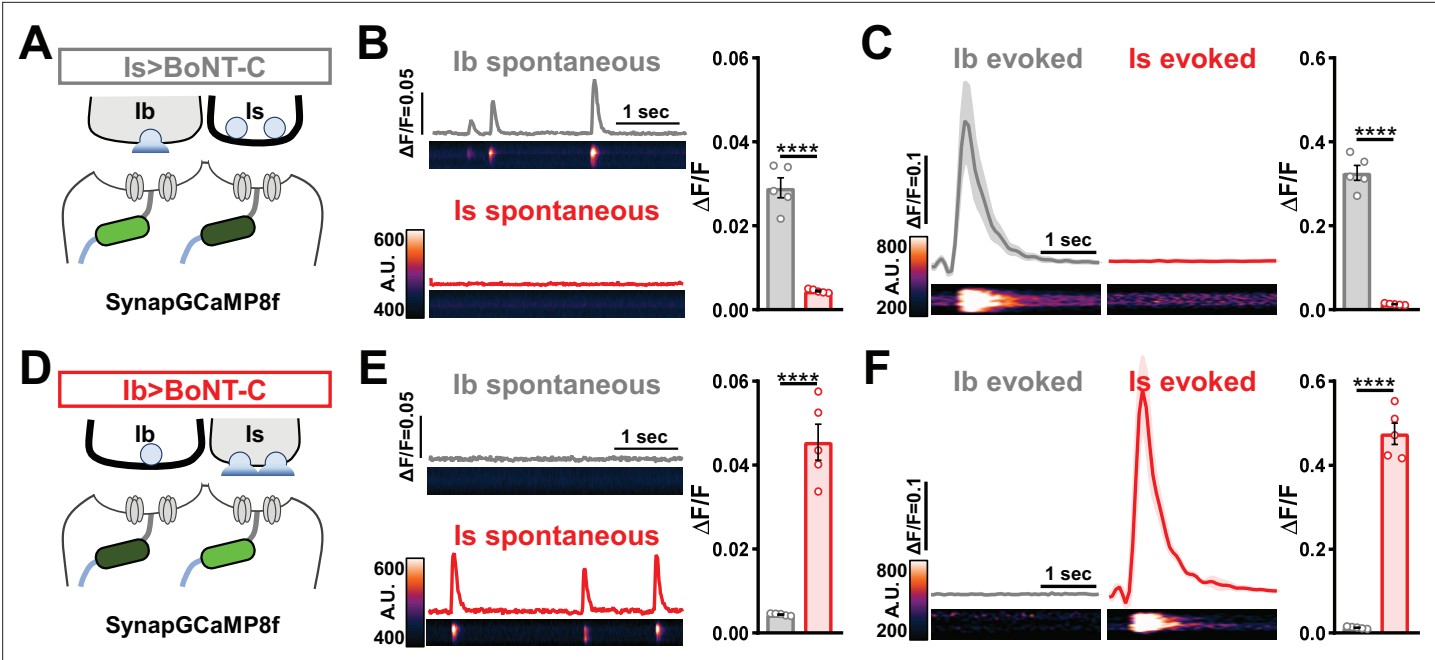

**Figure 5.** Postsynaptic Ca²⁺ signals are selectively eliminated by input-specific botulinum neurotoxin (BoNT-C) expression. (**A**) Schematic depicting silencing of MN-Is transmission by selective expression of BoNT-C in MN-Is. The postsynaptic Ca²⁺ indicator SynapGCaMP8f is indicated in the postsynaptic compartment, which should receive transmission only at MN-Ib neuromuscular junctions (NMJs). (**B**) Representative images and traces of spontaneous postsynaptic Ca²⁺ transients at MN-Ib and -Is NMJs using the new postsynaptic GCaMP8f indicator in Is>BoNT-C (*w;MHC>GCaMP8 f /+;R29E07-GAL4/UAS-BoNT-C*). Right: quantification of averaged transients, confirming that synaptic Ca²⁺ events are selectively eliminated at MN-Is NMJs but intact at MN-Ib. (**C**) Averaged traces and images of evoked postsynaptic Ca²⁺ transients at MN-Ib and -Is NMJs. Right: quantification of averaged evoked transients, confirming that evoked synaptic Ca²⁺ events are selectively eliminated at MN-Is NMJs but intact at MN-Ib. (**D**) Schematic depicting silencing of MN-Ib transmission by selective expression of BoNT-C in MN-Ib. (**E, F**) Similar analysis as shown in (**B, C**) at MN-Ib-silenced NMJs (Ib>BoNT-C; *w;MHC>GCaMP8 f /+;dHB9-GAL4/UAS-BoNT-C*). Synaptic Ca²⁺ events are selectively eliminated at MN-Ib NMJs but intact at MN-Is. Error bars indicate ± SEM. ****p<0.0001. Additional statistical details are summarized in ***Supplementary file 2***.

demonstrating no heterosynaptic structural plasticity is induced by input-specific BoNT-C expression. In contrast, TNT expression in MN-Is increased Is bouton numbers, as expected (see ***Figure 3***), but no changes in Ib boutons numbers were observed (***Figure 6A and C***), suggesting miniature-only transmission does not induce heterosynaptic structural changes. Finally, MN-Is ablation by rpr.hid expression fully ablated MN-Is inputs, while inducing a compensatory heterosynaptic increase in Ib bouton number (***Figure 6A and D***), as previously observed at other larval NMJs (***Aponte-Santiago et al., 2020***; ***Wang et al., 2021***). Parallel experiments driving BoNT-C, TNT, and rpr.hid transgenes in MN-Ib led to similar results as shown for MN-Is, with intrinsic but not heterosynaptic changes induced by *TNT* expression, ablation of both Is and Ib by rpr.hid expression, and no intrinsic or heterosynaptic structural plasticity induced by BoNT-C expression (***Figure 6F–I***).

It has been suggested that heterosynaptic plasticity may be distinctly expressed at different NMJs (***Wang et al., 2021***). We therefore examined synaptic growth at NMJs innervating muscles 12, 13, or 4 following TNT, rpr.hid, or BoNT-C expression in the Is MN innervating each of these muscles. However, intrinsic and heterosynaptic properties were similar at these other NMJs compared to our findings at muscle 6/7 (***Supplementary file 2***). Thus, while physical loss of MN-Is induced increased synaptic growth at MN-Ib, selective synaptic silencing by BoNT-C did not induce heterosynaptic structural plasticity. This reveals an important property about heterosynaptic structural plasticity at co-innervated muscles in *Drosophila*: no heterosynaptic structural plasticity is observed when the MN is physically present but functionally silent.

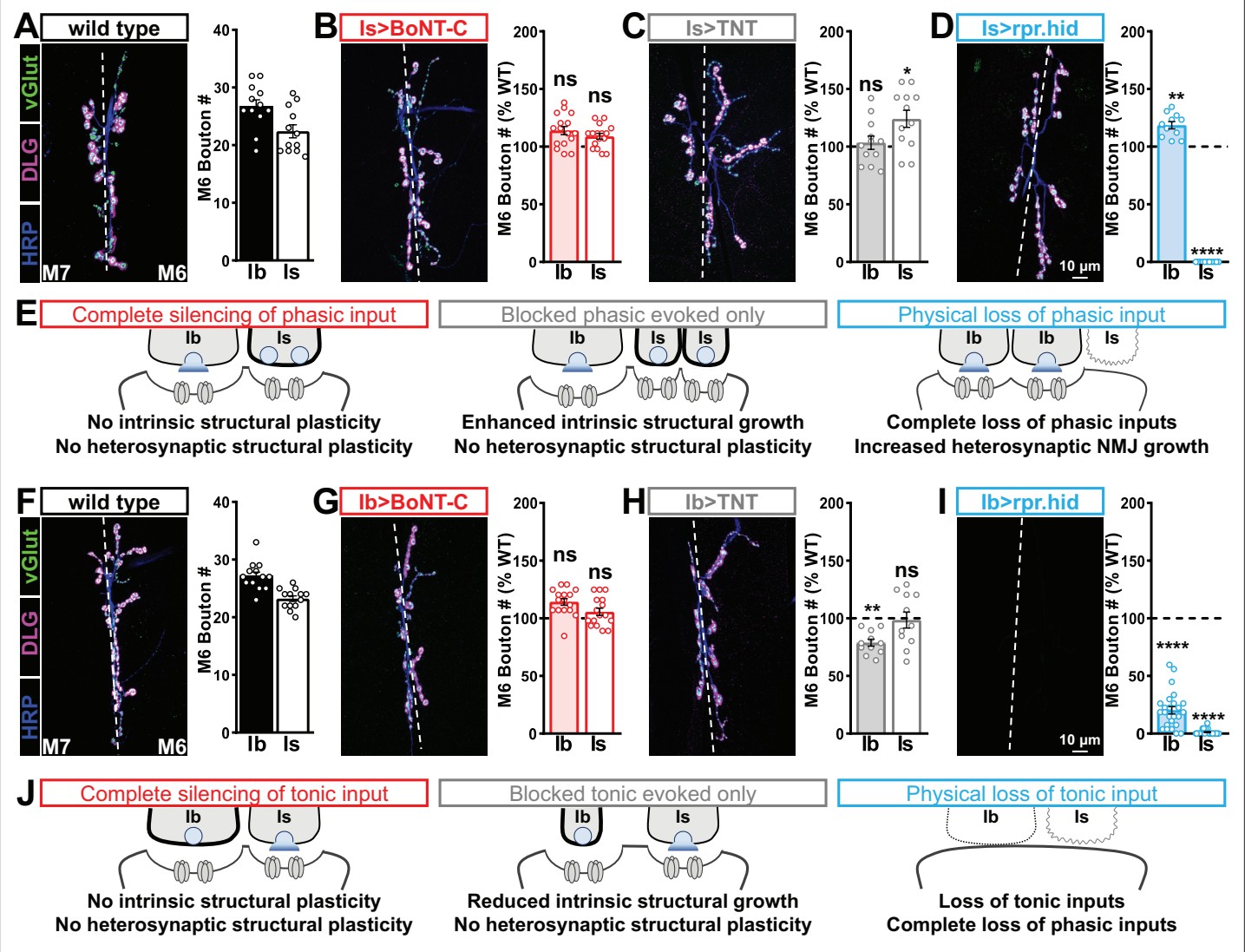

**Figure 6.** Selective silencing by botulinum neurotoxin (BoNT-C) does not induce heterosynaptic structural plasticity. (**A–D**) Representative neuromuscular junction (NMJ) images immunostained with anti-vGlut, -DLG, and -HRP in wild-type (**A**); selective silencing of MN-Is (Is>BoNT-C; *w;+;R29E07-GAL4/UAS-BoNT-C*) (**B**); tetanus toxin (TNT) expression in MN-Is (Is>TNT; *w;UAS-TNT/+;R29E07-GAL4/+*) (**C**); and ablation of MN-Is (Is>rpr.hid; *rpr.hid/+;R29E07-GAL4/+*) (**D**). Quantification of MN-Ib and MN-Is bouton number in wild-type and each condition normalized to wild-type values is shown on the right. Note that while no heterosynaptic structural plasticity is observed in bouton number Is>BoNT-C and Is>TNT, a compensatory heterosynaptic increase is found in Is>rpr.hid. (**E**) Schematic summarizing the results of (**A–D**). (**F–I**) Similar NMJ images and quantification as shown in (**A–D**) but with MN-Ib expression of BoNT-C (Ib>BoNT-C; *w;+;dHB9-GAL4/UAS-BoNT-C*), -TNT (Ib>TNT; *w;UAS-TNT/+;dHB9-GAL4/+*), or -rpr.hid (Ib>rpr.hid; *rpr.hid/+;dHB9-GAL4/+*). (**J**) Schematic summarizing the results of (**F–I**). Error bars indicate ± SEM. ****p<0.0001; ***p<0.001; *p<0.05; ns, not significant. Absolute values for normalized data and additional statistical details are summarized in ***Supplementary file 2***.

## Selective silencing by BoNT-C fully reconstitutes wild-type NMJ physiology

Complete neurotransmission from MN-Is and MN-Ib has never been electrophysiologically separated. Although input-specific silencing by BoNT-C does not induce heterosynaptic structural plasticity, it is possible that heterosynaptic functional adaptations might be imparted. Alternatively, input-specific silencing by BoNT-C may not induce functional changes in neurotransmission at the convergent input. We sought to distinguish between these possibilities. In standard electrophysiological recordings from wild-type NMJs, miniature transmission is an undefined blend of spontaneous events originating from both MN-Is and MN-Ib inputs, and evoked amplitudes reflect a nebulous composite of transmission from both motor inputs that does not reflect the behavior of any synapse that actually exists. We

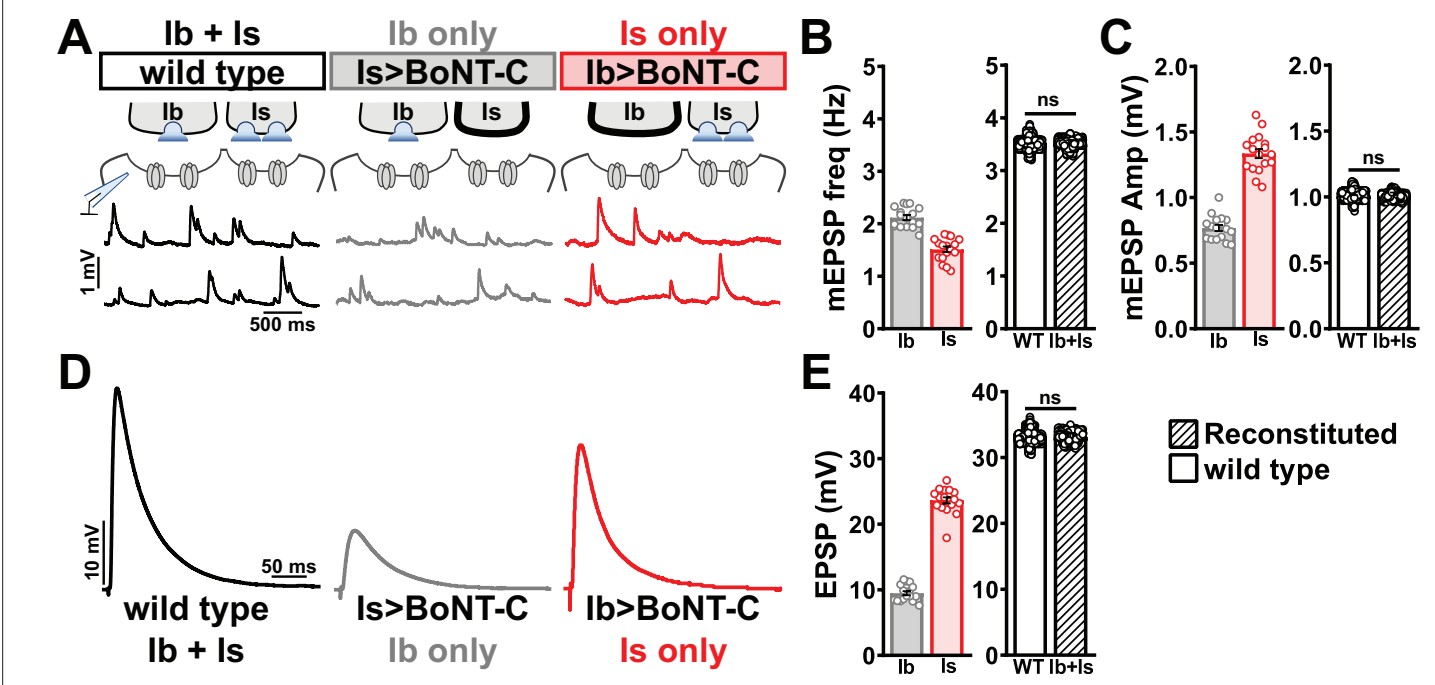

**Figure 7.** Selective silencing by botulinum neurotoxin (BoNT-C) fully reconstitutes wild-type neuromuscular junction (NMJ) physiology. (**A**) Schematic and representative miniature excitatory postsynaptic potential (mEPSP) traces of recordings from wild-type ('Ib+Is') or input-specific silencing of MN-Is ('Ib only') or MN-Ib ('Is only') by BoNT-C expression. (**B**) Quantification of mEPSP frequency in Ib only and Is only. A simple addition of these values (reconstituted) recapitulates the observed blended values observed in wild-type. (**C**) Quantification of mEPSP amplitude in Ib only and Is only. Note the substantial input-specific difference in mEPSP amplitude. A weighted average of these values from Ib and Is (reconstituted) fully recapitulates the average mEPSP amplitude values observed from recordings of wild-type NMJs. Reconstituted data sets were acquired by bootstrapping and resampling from the seed data sets (see 'Materials and methods' for additional details). (**D**) Representative EPSP traces from wild-type, Ib-only, and Is-only NMJs. (**E**) Quantification of EPSP amplitude from Ib-only and Is-only NMJs. A simple addition of these values (reconstituted) fully recapitulates the blended EPSP values obtained from wild-type recordings. Error bars indicate ± SEM. ns, not significant. Additional statistical details are summarized in *Supplementary file 2*.

The online version of this article includes the following figure supplement(s) for figure 7:

**Figure supplement 1.** Selective silencing by botulinum neurotoxin (BoNT-C) fully reconstitutes wild-type neuromuscular junction (NMJ) physiology using two-electrode voltage clamp.

reasoned that if wild-type physiology could be fully reconstituted after electrophysiological isolation of both miniature and evoked transmission from MN-Is and MN-Ib, this would confirm that no heterosynaptic functional plasticity is induced by input-specific BoNT-C expression. However, if wild-type physiology was not fully reconstituted, this would indicate that heterosynaptic functional plasticity was provoked by input-specific BoNT-C silencing.

We first compared miniature transmission at wild-type (Ib+Is), Ib only (Is>BoNT-C), and Is only (Ib>BoNT-C) NMJs. As expected, mEPSP frequency was reduced at both Ib and Is only compared to wild-type (*Figure 7A and B*). Importantly, substantial differences in mEPSP amplitude were observed at Is vs. Ib, where miniature events were over 70% increased at MN-Is NMJs compared to MN-Ib (1.34 mV ± 0.04 vs. 0.77 mV ± 0.02; *Figure 7A and C*), as predicted by previous studies (*Karunanithi et al., 2002*; *Newman et al., 2017*). This highlights the inaccurate quantal sizes reported in previous studies where transmission from each input was blended, and averaged miniature events were used to estimate quantal content at composite (Ib+Is) evoked amplitudes or with selective optogenetic stimulation (*Genç and Davis, 2019*; *Sauvola et al., 2021*). To determine whether wild-type miniature physiology could be reconstituted using the input-specific BoNT-C data, we first summed the Ib-only and Is-only miniature frequencies together, which resulted in a 3.54 Hz frequency that was statistically indistinguishable to the wild-type value (*Figure 7B*). Similarly, we combined the weighted average of mEPSP amplitudes from Ib-only and Is-only events, which provided a value of 1.00 mV ± 0.01, statistically similar to the wild-type value (0.998 ± 0.001; *Figure 7C*). Thus, a reductionist analysis of miniature

transmission from selectively silenced MN-Ib and MN-Is NMJs recapitulates the physiology observed at co-innervated muscles.

Next, we examined evoked EPSP events from composite (Ib+Is) or isolated (Ib or Is only) NMJs. At 0.4 mM $Ca^{2+}$ saline, evoked transmission from MN-Is is over twofold higher than MN-Ib (23.65 ± 0.48 vs. 9.50 ± 0.30; *Figure 7D and E*). When we summed EPSP amplitude from paired MN-Ib and -Is inputs, we obtained EPSP values not statistically different from the composite values (33.10 ± 0.03 vs. 32.60 ± 2.19; *Figure 7D and E*), suggesting that no heterosynaptic functional plasticity in evoked physiology is induced by selective BoNT-C expression. Finally, we were able to obtain an accurate quantal content value of synaptic vesicle release at MN-Ib and MN-Is, extracted from the disambiguated miniature and evoked transmission. Given the large quantal size of transmission from MN-Is, due to enhanced vesicle size (*Karunanithi et al., 2002*; *Newman et al., 2017*), less quantal content was observed than would be expected from the averaged quantal size, while the converse is true for transmission from MN-Ib (*Supplementary file 2*). To control for possible nonlinear summation effects, recordings in two-electrode voltage-clamp configuration identified similar results as compared with sharp electrode current clamp recordings (*Figure 7—figure supplement 1*). Together, these data demonstrate that composite transmission from co-innervated muscles can be fully reconstituted from isolated MN-Ib and MN-Is transmission enabled by selective BoNT-C transmission. Further, this finding underscores that no heterosynaptic structural or functional plasticity is induced when MN-Is or MN-Ib innervation is physically present but functionally silent.

### Heterosynaptic functional plasticity is only induced by physical loss of the convergent input

In our final set of experiments, we sought to clarify whether heterosynaptic functional plasticity, like structural plasticity, could be induced through selective expression of TNT or rpr.hid. Previous studies have suggested possible changes in neurotransmission (*Aponte-Santiago et al., 2020*; *Wang et al., 2021*), but the inability to isolate input-specific baseline miniature and evoked transmission obscured to what extent heterosynaptic functional plasticity was induced. Using selective expression of BoNT-C to isolate baseline MN-Ib or MN-Is transmission, we established baseline synaptic function from isolated MN-Ib (*Figure 8A*; 'Is silenced'; Is>BoNT-C) or MN-Is (*Figure 8C*; 'Ib silenced'; Ib>BoNT-C). Like heterosynaptic structural plasticity, selective expression of TNT in either MN-Is or MN-Ib did not induce heterosynaptic functional evoked plasticity (*Supplementary file 2*), although miniature events were confounded by their persistence after TNT expression (*Figure 1*). However, while ablation of MN-Is by rpr.hid expression eliminated mEPSP events from Is, mirroring Is>BoNT-C values, an adaptive enhancement in heterosynaptic evoked transmission was observed (*Figure 8B*). As expected, selective ablation of MN-Ib by Ib>rpr.hid expression eliminated most transmission (*Figure 8D*) due to loss of both MN-Ib and -Is innervation (*Figure 1—figure supplement 5*).

Together, these results illustrate important heterosynaptic plasticity rules between tonic and phasic neurons (*Figure 8* schematics). When neurotransmission is silenced from one input, no heterosynaptic structural or functional plasticity is induced. However, physical loss of one input provokes either an adaptive increase in synaptic strength due to enhanced synapse number and neurotransmitter release from the convergent input or loss of convergent innervation.

## Discussion

By screening a variety of botulinum neurotoxins, we identified BoNT-C to silence all neurotransmission when expressed in *Drosophila* MNs. Crucially, BoNT-C expression does not impair NMJ growth or synaptic structure and can isolate neurotransmission from phasic MN-Is and tonic MN-Ib when selectively expressed in either MN. Finally, no heterosynaptic structural or functional plasticity is induced at the convergent input following selective BoNT-C expression, revealing that heterosynaptic adaptive plasticity requires physical loss of the motor input. Thus, BoNT-C provides a powerful new approach to accurately disambiguate tonic vs. phasic neurotransmission at the *Drosophila* NMJ and provides a foundation from which to understand the induction mechanisms of heterosynaptic plasticity at this model glutamatergic synapse.

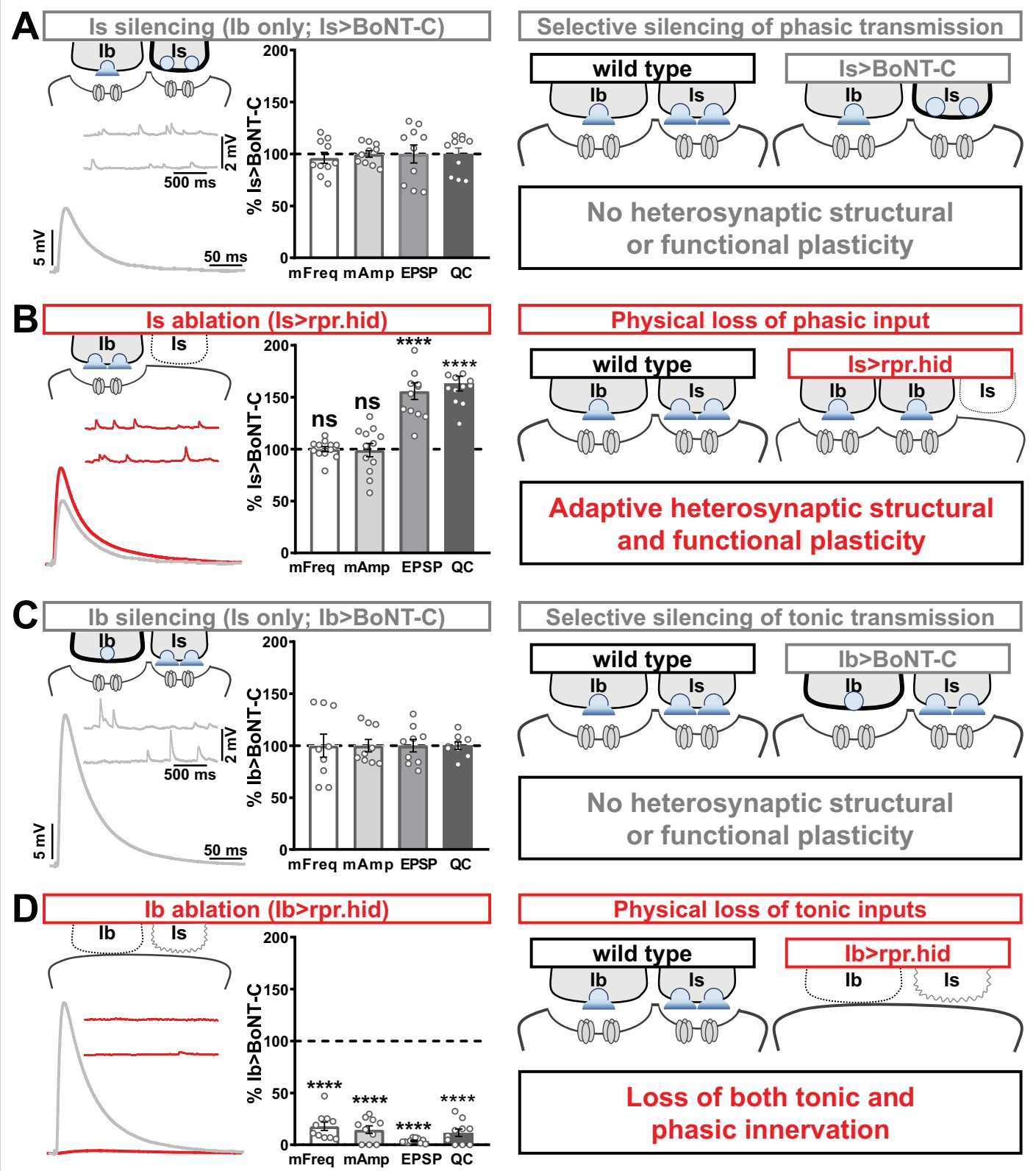

**Figure 8.** Selective silencing by botulinum neurotoxin (BoNT-C) does not induce heterosynaptic functional plasticity. (**A**) Schematic and representative electrophysiological traces of Is-silenced neuromuscular junctions (NMJs) (Ib only; Is>BoNT-C) as a baseline for isolated synaptic transmission from MN-Ib. Electrophysiological data (miniature excitatory postsynaptic potential [mEPSP] frequency, mEPSP amplitude, excitatory postsynaptic potential [EPSP] amplitude, and quantal content) are normalized to this same genotype (Is>BoNT-C). Right: schematic illustrating no heterosynaptic structural

*Figure 8 continued on next page*

*Figure 8 continued*

or functional plasticity is induced. (**B**) Schematic, traces, and quantification of recordings from NMJs in which MN-Is is ablated (Is<rpr.hid), normalized to baseline (Is>BoNT-C) values. Note that no differences in mEPSP values are observed following MN-Is ablation, while an apparent adaptive heterosynaptic increase in EPSP amplitude and quantal content is observed from the remaining MN-Ib, summarized at right. (**C**) Schematic, traces, and quantification of recordings from Ib-silenced NMJs (Is only; Ib>BoNT-C) as a baseline for isolated synaptic transmission from MN-Is. Electrophysiological data are normalized to this same genotype (Ib>BoNT-C). Right: schematic illustrating no heterosynaptic structural or functional plasticity is induced. (**D**) Schematic, traces, and quantification of recordings from NMJs in which MN-Ib are ablated (Ib<rpr.hid), normalized to baseline (Ib>BoNT C) values. Note that in this case synaptic transmission is essentially eliminated due to ablation of both MN-Ib and -Is inputs, summarized at right. Thus, when transmission is silenced by BoNT-C but both motor inputs are physically present, no heterosynaptic structural or functional plasticity is elicited, while when one of the motor inputs is physically absent, either adaptive heterosynaptic structural and functional plasticity or loss of the convergent input is observed. Error bars indicate ± SEM. ****p<0.0001; ***p<0.001; ns, not significant. Absolute values for normalized data and additional statistical details are summarized in *Supplementary file 2*.

## BoNT-C as a tool for effectively silencing neurotransmission

Neurotoxins that target SNARE complexes have been utilized in neuroscience research for decades. In *Drosophila*, transgenic control of TNT light chain expression was established decades ago (*Sweeney et al., 1995*) and widely used since to inhibit transmission in both peripheral and central neurons. However, while TNT expression eliminates evoked neurotransmission, miniature transmission persists (*Baines et al., 2001*; *Choi et al., 2014*; *Goel and Dickman, 2018*), as also observed in *n-syb* mutations (*Deitcher et al., 1998*), the target of TNT. This persistence of miniature transmission following TNT intoxication of MNs was exploited to explore the functions of spontaneous neurotransmission in synaptic growth and development at the *Drosophila* NMJ (*Choi et al., 2014*) and the proportion of postsynaptic $Ca^{2+}$ influx driven by evoked released at tonic vs. phasic NMJs (*Newman et al., 2017*). However, despite the identification of GAL4 drivers specific for tonic vs. phasic MNs, two major limitations rendered TNT a suboptimal approach to isolate input-specific transmission at the fly NMJ. First, the persistence of miniature transmission obscures the accurate determination of accurate quantal content values from either MN, particularly given the large difference in quantal size between the two inputs (*Figure 7*). In addition, TNT expression alters synaptic growth at MN terminals (*Figure 3*; *Goel and Dickman, 2018*), and heterosynaptic plasticity was reported with input-specific TNT expression (*Aponte-Santiago et al., 2020*). Interestingly, while we did not observe heterosynaptic structural or functional plasticity following input-specific TNT expression at the muscle 6/7 NMJ, accurate evoked EPSP amplitudes were observed (*Figure 8*), suggesting that input-specific TNT expression can alter synaptic growth and block evoked transmission without inducing any apparent changes at the convergent input.

The isolation of BoNT-C was fortunate given the variable activity and toxicity previously reported with BoNTs. In the *Drosophila* visual system, BoNT transgene expression led to variable substrate cleavage where, for example, BoNT-A only partially cleaved SNAP-25 and Syntaxin (*Backhaus et al., 2016*). Similarly, we also observed variable activity between independent transgenic inserts of BoNT-E, where expression of some transgenes appeared to make animals less healthy than others (see 'Materials and methods'). In addition, expression of TNT or BoNT can lead to neurodegeneration (*Berliocchi et al., 2005*; *Peng et al., 2013*; *Zhao et al., 2010*). Although we cannot rule out that BoNT-C may induce toxicity when expressed in other cell types or for longer durations in MNs, we find no evidence for toxicity or altered health when BoNT-C is expressed in MNs throughout larval development. This is likely due to BoNT-C exhibiting specific cleavage activity for *Drosophila* Syntaxin (*Backhaus et al., 2016*) in synaptic SNARE complexes as well as a fortuitous genomic insertion that ensures moderate expression of the transgene. BoNT-C, when expressed in MNs, appears to selectively target Syx in complexes controlling regulated synaptic vesicle fusion at synapses, distinct from complexes involved in constitutive post-Golgi membrane trafficking. BoNT-C expression in muscle also leads to lethality (*Supplementary file 1*; *Backhaus et al., 2016*), so it is likely that BoNT-C can target Syx in complexes with post-Golgi SNARE machinery in non-neuronal cell types. There are 10 *Drosophila syntaxin* genes and three Syx1A isoforms, so BoNT-C may preferentially target subsets of specific Syx complexes depending on the tissue it is expressed.

Several important properties of synaptic structure and function in tonic vs. phasic MNs have been confirmed and extended by selective BoNT-C silencing. First, phasic NMJs are responsible for about twice the postsynaptic depolarization compared to tonic, at least with regard to single stimuli under

the conditions used in our study. However, despite this obvious difference in synaptic strength, quantal content values released between MN-Is vs. -Ib are more similar than would be apparent based on the substantial difference in quantal size. Second, tonic vs. phasic firing patterns do not determine the specialized abundance and localization of postsynaptic glutamate-receptive fields. Third, synaptic morphogenesis and structure does not require vesicular neurotransmitter release at glutamatergic synapses as BoNT-C NMJs appear indistinguishable from wild-type. Although several studies have shown neurotransmitter release is not necessary for the initial establishment of synapses (*Banerjee et al., 2022*; *Sando et al., 2017*; *Sigler et al., 2017*; *Varoqueaux et al., 2002*), whether miniature transmission was necessary for clustering of glutamate receptors at the fly NMJ has provoked some controversy (*Otsu and Murphy, 2003*; *Saitoe et al., 2002*; *Verstreken and Bellen, 2002*). It will be of significant interest in future studies to now determine the full electrophysiological properties of tonic vs. phasic neurons, including vesicle pools, short-term plasticity, and release probability across various extracellular $Ca^{2+}$ conditions. Furthermore, to what extent synaptic components exhibit specialized functions at tonic vs. phasic MNs, as was recently shown for *tomosyn* (*Sauvola et al., 2021*), will also be an exciting area to leverage BoNT-C silencing going forward. Input-specific silencing by BoNT-C now provides a foundation to elucidate fundamental electrophysiological differences between tonic and phasic neuronal subtypes and, importantly, to re-evaluate specialized functions of *Drosophila* synaptic genes in which initial characterizations relied on blended transmission from both inputs.

## Insights into synapse development and heterosynaptic plasticity revealed by BoNT-C

Physical loss of tonic or phasic motor inputs appears to be the key inductive process capable of inducing adaptive heterosynaptic plasticity. Input-specific silencing of neurotransmission by BoNT-C, or even blockade of evoked release only by TNT, failed to elicit any structural or functional changes in the convergent input at the NMJs we assayed (muscle 6/7, 12/13, and 4). Genetic ablation of MN-Ib largely eliminated tonic inputs, as expected, and completely prevented phasic innervation at the convergent muscle 6/7 NMJ. This dramatic change contrasts with the more subtle differences in MN-Is innervation following MN-Ib ablation at the muscle 1 NMJ (*Aponte-Santiago et al., 2020*). Indeed, we found no evidence that phasic innervation ever occurred when tonic MNs were ablated, with no remnants of phasic postsynaptic structures remaining at the muscle target, while other muscle targets innervated by the same phasic input appeared largely unchanged. It appears MN-Is NMJs were never formed since no 'synaptic footprints' were observed, which are postsynaptic structures that remain after initial synaptogenesis and growth followed by MN retraction observed in neuromuscular degeneration (*Eaton et al., 2002*; *Perry et al., 2017*). The failure of phasic innervation following tonic ablation suggests that during development the tonic MN provides necessary 'instructive' cues for proper phasic innervation, akin to pioneer axons important for axon guidance (*Araújo and Tear, 2003*; *Raper and Mason, 2010*). Tonic MN-Ib serving as guidance cues for phasic innervation may make sense given that each muscle target receives tonic innervation from a single MN, while phasic MNs innervate groups of multiple muscles (*Hoang and Chiba, 2001*).

Conversely, ablation of phasic MN-Is inputs was the only condition in which we found evidence for adaptive structural and functional heterosynaptic plasticity at tonic MN-Ib inputs. It was in this condition where clear evidence for adaptive heterosynaptic structural plasticity was also reported in previous studies (*Aponte-Santiago et al., 2020*; *Wang et al., 2021*). Importantly, BoNT-C silencing establishes that loss of neurotransmission from the phasic input per se is not sufficient to induce any adaptive heterosynaptic plasticity. Rather, physical loss of the phasic input must provide an instructive cue, perhaps through signaling from peripheral glia and/or the common muscle target, that elicits adaptive plasticity at the remaining tonic input.

It would be appealing if loss of transmission from one input could induce a homeostatic adjustment at the convergent input that maintains stable postsynaptic excitation. Clearly this is not the case at the *Drosophila* NMJ. BoNT-C expression in the tonic input depresses transmission by ~1/3, while loss of phasic transmission reduces NMJ strength by ~2/3, with no evidence of heterosynaptic plasticity observed. In our study and in other cases where heterosynaptic plasticity has been reported (*Aponte-Santiago et al., 2020*; *Wang et al., 2021*), the seemingly adaptive changes in synaptic growth or function are quite subtle, nowhere near the levels that would be necessary to restore basal levels of transmission. For example, total bouton number remained highly reduced following phasic MN-Is

ablation (wild-type: 52 ± 1.5; Is>rpr.hid: 32 ± 0.8 bouton numbers), with a corresponding reduction in synaptic strength (wild-type: 32.6 ± 2.2 mV; Is>rpr.hid: 14.2 ± 0.7 mV EPSP values). However, while loss of transmission or innervation to a particular target muscle does not induce homeostatic plasticity, a converse manipulation, in which innervation is biased between adjacent muscle targets, does elicit two distinct forms of target-specific homeostatic plasticity that stabilizes synaptic strength (*Davis and Goodman, 1998*; *Goel et al., 2020*). Hyper-innervation of both tonic and phasic inputs on muscle 6 homeostatically reduces release probability from both inputs without any obvious postsynaptic changes (*Davis and Goodman, 1998 Goel et al., 2020*; *Goel et al., 2019b*). In contrast, hypo-innervation of both tonic and phasic inputs on the adjacent muscle 7 leads to no functional changes in the neurons, while a homeostatic enhancement in postsynaptic glutamate receptor abundance compensates for reduced transmitter release to restore synaptic strength (*Goel et al., 2020*). Illuminating the complex synaptic dialogue between pre- and postsynaptic compartments within and between common targets will clarify how adaptive and homeostatic plasticity mechanisms are engaged in at NMJs and in motor circuits in general.

# Materials and methods

## Fly stocks (*Pielage et al., 2005*)

Experimental flies were raised at 25°C on standard molasses food. The $w^{1118}$ strain was used as the wild-type control unless otherwise noted as this is the genetic background for which all genotypes are bred. For optogenetic experiments, flies were raised in consistent dark conditions on standard food supplemented with 500 µM all-trans-retinal (#R2500, Sigma-Aldrich). Second-instar larvae were then transferred to fresh food containing 500 µM all-trans-retinal. The following fly stocks were used: *OK6-GAL4* (*Aberle et al., 2002*), *UAS-rpr.hid* (*Zhou et al., 1997*), *OK319-GAL4* (*Sweeney et al., 1995*), *vGlut^{SS1}* (*Sherer et al., 2020*), *B3RT-vGlut-B3RT* (*Sherer et al., 2020*), and *UAS-B3* (*Sherer et al., 2020*). The following fly strains were generated in this study: *UAS-BoNT-A*, *UAS-BoNT-B*, *UAS-BoNT-C*, *UAS-BoNT-E*, and SynapGCaMP8f (*MHC-CD8-GCaMP8f-sh*). All other stocks were obtained from Bloomington Drosophila Stock Center (BDSC): *UAS-ChR2^{T159C}* (#58373), *dHb9-GAL4* (Ib-GAL4, #83004), *GMR27E09-GAL4* (Is-GAL4, #49227), *UAS-TNT* (#28838), $w^{1118}$ (#5905), *D42-GAL4* (#8816), and *UAS-CD4::tdGFP* (#35839). Details of these and additional stocks and their sources are listed in.

## Molecular biology

To generate UAS-BoNT-A, -B, - C, and -E transgenes, we cloned sequences encoding each BoNT light chain (from plasmids shared by Matt Kennedy, University of Colorado, USA) into the gateway vector donor plasmid (Thermo Fisher, #K240020). We then transferred these sequences into the final *pUASt* destination vector from Carnegie *Drosophila* Gateway Vector Collection using the Gateway reaction kit (Thermo Fisher, #11791020). To generate SynapGCaMP8f (*MHC-CD8-GCaMP8f-Sh*), we obtained the SynapGCaMP6f transgenic construct (*Newman et al., 2017*) and replaced the sequence encoding GCaMP6f with a sequence encoding GCaMP8f (#162379, Addgene) using Gibson assembly. Transgenic stocks were inserted into $w^{1118}$ by Bestgene, Inc (Chino Hills, CA) using P-element-mediated random insertion and subsequently mapped and balanced. Pilot crosses of various BoNT transgenes to various neural GAL4 drivers were first used to select the inserts that appeared to induce the most consistent lethality and used for further analysis. Variable toxicity was noted in the BoNT-A and BoNT-E

lines, where some inserts led to lethality when crossed pan-neural drivers, while other inserts were viable.

## Electrophysiology

Electrophysiological recordings were performed as described (*Kiragasi et al., 2020*; *Li et al., 2021*) using modified hemolymph-like saline (HL-3) containing 70 mM NaCl, 5 mM KCl, 10 mM MgCl$_2$, 10 mM NaHCO$_3$, 115 mM sucrose, 5 mM trehalose, 5 mM HEPES, and 0.5 mM CaCl$_2$, pH 7.2, from cells with resting potentials between –60 and –75 mV and input resistances >6 MΩ. Recordings were performed on an Olympus BX61 WI microscope using a ×40/0.80 NA water-dipping objective and acquired using an Axoclamp 900A amplifier, Digidata 1440A acquisition system, and pClamp 10.5 software (Molecular Devices). mEPSPs were recorded in the absence of any stimulation. EPSPs were recorded by delivering 20 electrical stimulations at 0.5 Hz with 0.5 ms duration to MNs using an ISO-Flex stimulus isolator (A.M.P.I.) with stimulus intensities set to avoid multiple EPSPs. All recordings were made on abdominal muscle 6 in segments A2 or A3 of third-instar larvae of both sexes. Data were analyzed using Clampfit (Molecular Devices), Mini Analysis (Synaptosoft), or Excel (Microsoft). Averaged mEPSP amplitude, mEPSP frequency, EPSP amplitude, and quantal content values were calculated for each genotype. Simulated data in *Figure 7* of WT, Is, and Ib were acquired by boot-strapping and resampling. EPSP, mEPSP, and mEPSP frequency data were resampled 1000 times from the raw seed data set shown in the left panels in *Figure 7B, C and E* and *Supplementary file 1*. 1000 mean values were calculated from all the resampled data sets, followed by calculating EPSP amplitude, mEPSP frequency, and mEPSP amplitude for paired Is+Ib using the following equation: $EPSP_{(Is+Ib)} = EPSP_{(Is)} + EPSP_{(Ib)}$; $freq._{(Is+Ib)} = freq._{(Is)} + freq._{(Is)}$; $mEPSP_{(Is+Ib)} = (mEPSP_{(Is)} \times freq._{(Is)} + mEPSP_{(Ib)} \times freq._{(Is)})/freq._{(Is+Ib)}$.

## Ca²⁺ imaging and analysis

Third-instar larvae were dissected in ice-cold saline. Imaging was performed in modified HL-3 saline with 1.5 mM Ca$^{2+}$ added using a Zeiss Examiner A1 widefield microscope equipped with a ×63/1.0 NA water immersion objective. NMJs on muscle 6 were imaged at a frequency of 100 fps (512 × 256 pixels) with a 470 nm LED light source (Thorlabs) using a PCO sCMOS4.2 camera. Spontaneous Ca$^{2+}$ events were imaged at NMJs during 120 s imaging sessions from at least two different larvae. Evoked Ca$^{2+}$ events were induced by delivering 10 electrical stimulations at 0.5 Hz. Horizontal drifting was corrected using ImageJ plugins (*Li, 2008*) and imaging data with severe muscle movements were rejected as described (*Ding et al., 2019*). Three regions of interest (ROIs) were manually selected using the outer edge of terminal Ib boutons observed by baseline GCaMP signals with ImageJ (*Rueden et al., 2017*; *Schindelin et al., 2012*). Ib and Is boutons were defined by baseline GCaMP8f fluorescence levels, which are two- to threefold higher at Ib NMJs compared to their Is counterparts at a particular muscle. Fluorescence intensities were measured as the mean intensity of all pixels in each individual ROI. ΔF for a spontaneous event was calculated by subtracting the baseline GCaMP fluorescence level F from the peak intensity of the GCaMP signal during each spontaneous event at a particular bouton as previously detailed (*Li et al., 2021*). Baseline GCaMP fluorescence was defined as the average fluorescence of 2 s in each ROI without spontaneous events. ΔF/F was calculated by normalizing ΔF to baseline signal F. For each ROI under consideration, the spontaneous event ΔF/F value was averaged for all events in the 60 s time range to obtain the mean quantal size for each bouton. First-instar larvae were prepared in ice-cold HL-3 saline with 1.5 mM Ca$^{2+}$ added without dissection. Spontaneous Ca$^{2+}$ events were acquired through cuticles of first-instar larvae with the same protocol described above. Data analysis was performed with customized Jupyter Note codes (*Source code 1* ).

## Optogenetics

Electrophysiology recording with optogenetics stimulation were performed with the same set up detailed above with a ×40/1.0 NA water immersion objective. Dissection and electrophysiological recordings were performed in the same modified HL-3 saline described above. To stimulate EPSP events, 0.5 ms light pulses of 470 nm were delivered from an LED driver (Thorlabs) at 0.5 Hz, triggered by a TTL pulse driven from a Digidata 1440 DAC programmed using pClamp 10.5 software (Molecular

Devices). Light intensity was controlled by the LED driver to avoid multiple EPSP events. The power range were calibrated between 5 and 20 mW/cm$^2$ under the objective.

## Immunocytochemistry

Larvae were dissected in ice-cold 0 Ca$^{2+}$ HL-3 and immunostained as described (*Kiragasi et al., 2017*). In brief, larvae were either fixed in Bouin's fixative for 5 min (Sigma, HT10132-1L), 100% ice-cold ethanol for 5 min, or 4% paraformaldehyde (PFA) for 10 min. Larvae were then washed with PBS containing 0.1% Triton X-100 (PBST) for 30 min, blocked with 5% Normal Donkey Serum followed by overnight incubation in primary antibodies at 4°C. Preparations were then washed 3× in PBST, incubated in secondary antibodies for 2 hr, washed 3× in PBST, and equilibrated in 70% glycerol. Prior to imaging, samples were mounted in VectaShield (Vector Laboratories). Details of all antibodies, their source, dilution used, and references are listed in .

## Confocal imaging and analysis

Samples were imaged as described (*Kikuma et al., 2019*) using a Nikon A1R Resonant Scanning Confocal microscope equipped with NIS Elements software and a ×100 APO 1.4NA oil immersion objective using separate channels with four laser lines (405 nm, 488 nm, 561 nm, and 647 nm). For fluorescence intensity quantifications of BRP, vGlut, GluRIIA, GluRIIB, and GluRIID, z-stacks were obtained on the same day using identical gain and laser power settings with z-axis spacing between 0.15 and 0.20 µm for all genotypes within an individual experiment. Maximum intensity projections were utilized for quantitative image analysis using the general analysis toolkit of NIS Elements software. The fluorescence intensity levels of BRP, vGlut, GluRIIA, GluRIIB, and GluRIID immunostaining were quantified by applying intensity thresholds and filters to binary layers in the 405 nm, 488 nm, or 561 nm channels. The mean intensity for each channel was quantified by obtaining the average total fluorescence signal for each individual punctum and dividing this value by the puncta area. The mean area of each GluR puncta was measured and defined as GluR puncta size. A mask was created around the HRP channel, used to define the neuronal membrane, and only puncta within this mask were analyzed to eliminate background signals. Boutons were defined as vGlut puncta, and DLG co-staining was used to define boutons at MN-Is vs. -Ib NMJs. NMJ area was measured as the sum area of all boutons labeled by HRP. All measurements based on confocal images were taken from synapses acquired from at least six different animals.

## Electron microscopy

EM analysis was performed as described previously (*Goel et al., 2019a*). Wandering third-instar larvae were dissected in Ca$^{2+}$-free HL-3 and then fixed in 2.5% glutaraldehyde/0.1 M cacodylate buffer at 4°C. Larvae were then washed three times for 20 min in 0.1 M cacodylate buffer. The larval pelts were then placed in 1% osmium tetroxide/0.1 M cacodylate buffer for 1 hr at room temperature. After washing the larva twice with cacodylate and twice with water, larvae were then dehydrated in ethanol. Samples were cleared in propylene oxide and infiltrated with 50% Eponate 12 in propylene oxide overnight. The following day, samples were embedded in fresh Eponate 12. EM sections were obtained on a Morgagni 268 transmission electron microscope (FEI). NMJs were serial sectioned at a 60–70 nm thickness. The sections were mounted on Formvar-coated single-slot grids and viewed at a 23,000 magnification and were recorded with a Megaview II CCD camera. Images were analyzed using the general analysis toolkit in the NIS Elements software and ImageJ software. Active zone area was measured by defining a circle area with a diameter of the length of the active zone dense at the center of T-bar structure. Synaptic vesicle density was analyzed by vesicle numbers normalized to active zone area.

## Statistical analysis

Data were analyzed using GraphPad Prism (version 7.0) or Microsoft Excel software (version 16.22). Sample values were tested for normality using the D'Agostino & Pearson omnibus normality test, which determined that the assumption of normality of the sample distribution was not violated. Data were then compared using either a one-way ANOVA and tested for significance using a Tukey's multiple comparison test or using an unpaired two-tailed Student's *t*-test with Welch's correction. In all figures, error bars indicate ± SEM, with the following statistical significance: *p<0.05, **p<0.01,

***p<0.001, ****p<0.0001; ns, not significant. Additional statistics and sample number n values for all experiments are summarized in *Supplementary file 1* and *Supplementary file 2*.

## Acknowledgements

We thank Matt Kennedy (University of Colorado, Aurora, CO, USA) for sharing botulinum neurotoxin plasmids, Zachary Newman and Udi Issacof (UC Berkeley, Berkeley, CA, USA) for sharing the SynapGCaMP6f plasmid, Steve Stowers (Montana State University, Bozeman, MT, USA) for sharing the vGlut conditional knock out lines, and Greg Macleod (Florida Atlantic University, Jupiter, FL, USA) for important discussions on MN-Is and MN-Ib physiology and for comments on an earlier version of this manuscript. We acknowledge the Developmental Studies Hybridoma Bank (Iowa City, IA, USA) for antibodies used in this study and the Bloomington Drosophila Stock Center for fly stocks (NIH P40OD018537). We thank Landon Porter and Brian Leung for technical assistance in validating reagents. This work was supported by grants from the National Institutes of Health (NS091546 and NS111414) to DD.

## Additional information

### Competing interests

Dion Dickman: Reviewing editor, eLife. The other authors declare that no competing interests exist.

### Funding

| Funder | Grant reference number | Author |
| --- | --- | --- |
| National Institutes of Health | NS091546 | Dion Dickman |
| National Institutes of Health | NS111414 | Dion Dickman |

The funders had no role in study design, data collection and interpretation, or the decision to submit the work for publication.

### Author contributions

Yifu Han, Data curation, Formal analysis, Investigation, Methodology, Writing – review and editing; Chun Chien, Data curation, Investigation; Pragya Goel, Conceptualization, Data curation, Formal analysis, Validation, Investigation, Methodology; Kaikai He, Data curation, Validation; Cristian Pinales, Data curation; Christopher Buser, Supervision, Methodology; Dion Dickman, Conceptualization, Supervision, Funding acquisition, Investigation, Methodology, Writing – original draft, Project administration, Writing – review and editing

### Author ORCIDs

Yifu Han http://orcid.org/0000-0002-1201-654X
Cristian Pinales http://orcid.org/0000-0002-0826-5308
Christopher Buser http://orcid.org/0000-0002-4379-3878
Dion Dickman http://orcid.org/0000-0003-1884-284X

### Decision letter and Author response

Decision letter https://doi.org/10.7554/eLife.77924.sa1
Author response https://doi.org/10.7554/eLife.77924.sa2

## Additional files

### Supplementary files

- Supplementary file 1. Characterization of BoNT-A, BoNT-B, and BoNT-E.
- Supplementary file 2. Absolute values for normalized data and complete statistical details.
- Transparent reporting form

• Source code 1. Customized Jupyter Note codes.

## Data availability

All data generated or analyzed during this study are included in the manuscript and supporting files. In particular, full details of the data are included in Supplementary files 1 and 2.

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

# Appendix 1

## Appendix 1—key resources table

| Reagent type (species) or resource | Designation | Source or reference | Identifiers | Additional information |
|---|---|---|---|---|
| Antibody | Mouse monoclonal | Developmental Studies Hybridoma Bank (DSHB) | AB_528269 | 1:50 |
| Antibody | Mouse monoclonal | DSHB | AB_2314866 | 1:100 |
| Antibody | Mouse monoclonal | DSHB | AB_528203 | 1:100 |
| Antibody | Mouse monoclonal | DSHB | AB_2617422 | 1:500 |
| Antibody | Rabbit polyclonal | *Pielage et al., 2005* | | 1:10000 |
| Antibody | Rabbit polyclonal | *Perry et al., 2017* | | 1:1000 |
| Antibody | Guinea polyclonal | *Goel and Dickman, 2018* | | 1:2000 |
| Antibody | Guinea polyclonal | *Perry et al., 2017* | | 1:1000 |
| Antibody | Alexa Fluor 647 conjugated Goat anti-Horseradish Peroxidase | Jackson ImmunoResearch Laboratories (Jackson) | 123-605-021 | 1:400 |
| Antibody | Alexa Fluor 488 conjugated secondary antibodies | Jackson | 706-545-148, 715-545-150, 711-545-152 | 1:400 |
| Antibody | Cy3-conjugated secondary antibodies | Jackson | 706-165-148, 715-165-150, 711-165-152 | 1:400 |
| Antibody | DyLight 405-conjugated secondary antibodies | Jackson | 706-475-148, 715-475-150 | 1:400 |
| Antibody | Alexa Fluor 647 conjugated Goat anti-Phalloidin | ThermoFisher | A22287 | 1:1000 |
| Genetic reagent (*D. melanogaster*) | UAS-BoNT-C | This study | | See Materials and methods, subsection Molecular biology. |
| Genetic reagent (*D. melanogaster*) | UAS-BoNT-A | This study | | Same as above |
| Genetic reagent (*D. melanogaster*) | UAS-BoNT-B | This study | | Same as above |
| Genetic reagent (*D. melanogaster*) | UAS-BoNT-E | This study | | Same as above |
| Genetic reagent (*D. melanogaster*) | MHC-CD8-GCaMP8f-Sh (SynapGCaMP8f) | This study | | Same as above |
| Genetic reagent (*D. melanogaster*) | OK6-GAL4 | *Aberle et al., 2002* | | |
| Genetic reagent (*D. melanogaster*) | UAS-rpr.hid | *Zhou et al., 1997* | | |
| Genetic reagent (*D. melanogaster*) | OK319-GAL4 | *Sweeney et al., 1995* | | |
| Genetic reagent (*D. melanogaster*) | vGlut$^{SS1}$ | *Sherer et al., 2020* | | |
| Genetic reagent (*D. melanogaster*) | B3RT-vGlut-B3RT | *Sherer et al., 2020* | | |
| Genetic reagent (*D. melanogaster*) | UAS-B3 | *Sherer et al., 2020* | | |

*Appendix 1 Continued on next page*

*Appendix 1 Continued*

| Reagent type (species) or resource | Designation | Source or reference | Identifiers | Additional information |
|---|---|---|---|---|
| Genetic reagent (*D. melanogaster*) | UAS-ChR2$^{T159C}$ | Bloomington *Drosophila* Stock Center (BDSC) | 58373 | |
| Genetic reagent (*D. melanogaster*) | UAS-CD4::tdGFP | BDSC | 35839 | |
| Genetic reagent (*D. melanogaster*) | dHb9-GAL4 | BDSC | 83004 | |
| Genetic reagent (*D. melanogaster*) | GMR27E09-GAL4 | BDSC | 49227 | |
| Genetic reagent (*D. melanogaster*) | UAS-TNT | BDSC | 28838 | |
| Genetic reagent (*D. melanogaster*) | $w^{1118}$ | BDSC | 5905 | |
| Genetic reagent (*D. melanogaster*) | D42-GAL4 | BDSC | 8816 | |
| Recombinant DNA reagent | pUASt destination vector | *Drosophila* Genetics Resource Center (DGRC) | 1129 | |
| Chemical, compound, drug | all trans-Retinal | Sigma-Aldrich | R2500 | |
| Software | NIS Elements software | Nikon | 4.51.01 | |
| Software | Axon pCLAMP Clampfit | Molecular Devices | 10.7 | |
| Software | MiniAnalysis | Synaptosoft | 6.0.3 | |
| Software | GraphPad Prism | GraphPad | 8.0.1 | |
| Software | Jupyter Notebook | Anaconda | 6.0.1 | |
| Software | ImageJ (Fiji) | **Rueden et al., 2017** | | |

