## [Editor Report]

This article reports a new genetically encoded neuronal silencer BoNT-C for use in *Drosophila*. The authors show that it fully blocks neurotransmission in two classes of *Drosophila* motor neurons (Is and Ib; tonic and phasic, respectively). They also update a GCaMP postsynaptic reporter SynapGCaMP8f. They selectively silence Ib or Is neurons to disambiguate the neurotransmission properties of each neuron, and finally, show that silencing either Ib or Is neurons does not induce heterosynaptic structural or functional plasticity. The data are convincing and the new silencing tool will be widely used.

---

## [Decision Letter]

**Decision letter after peer review:**

Thank you for submitting your article "Botulinum neurotoxin accurately separates tonic vs phasic transmission and reveals heterosynaptic plasticity rules in *Drosophila*" for consideration by *eLife*. Your article has been reviewed by 4 peer reviewers, including Chris Q Doe as Reviewing Editor and Reviewer #1, and the evaluation has been overseen by Lu Chen as the Senior Editor. The following individual involved in review of your submission has agreed to reveal their identity: Aref Arzan Zarin (Reviewer #2).

Essential revisions:

Note that this is a single integrated reviewer response taken from four independent reviewers.

Recommendations for Authors (Integrated review)

1. Determine precisely when BoNT-C blocks synaptic transmission. Is embryonic transmission completely blocked or does the blockage occur later in development? This is critical for interpretation of the dataset.

2. The authors should do a more careful developmental study of synapse maturation, particularly PSD development, to assay if the block in synaptic transmission (assuming it occurs early on) leads to delay in PSD maturation, or simply has no effect. I'm not sure the authors can make their conclusion so strongly given they only assayed the NMJ in late 3rd instars. Assaying PSD length and GluRIIA/B distribution in 1st and 2nd instars, where they would have more temporal control to examine activity-dependent PSD development, would be useful. Perhaps at earlier time points the authors would see a delay in PSD/GluR field maturation, but that this difference is less apparent at older synapses where delays might be masked. If so, this would indicate activity can control the rate of GluR field assembly, but is not required for the process to eventually complete.

3. Please discuss why the block in synaptic transmission does not trigger the PHP response the authors have previously described with enhanced BRP accumulation at presynaptic AZs.

4. Can the authors provide a discussion of why their BoNT-C does not block all post-Golgi traffic given Syx1 is required for both SV fusion and post-Golgi vesicle fusion with the plasma membrane.

5. The idea that 'composite values of Is and Ib neurotransmission can be fully recapitulated by isolated physiology from each input' (Figure 7) is not a new experiment and the idea of "reconstitution of wild-type NMJ physiology" by simply summing up evoked responses using sharp electrode recordings is questionable. Because they see a sum 33 mV EPSP and a IB of 9.5 mV and a Is of 23 mV doesn't really mean anything other than the Is has a stronger output, as has been previously shown by numerous labs. Without voltage clamp, it becomes harder and harder to depolarize the muscle as you get further from the resting potential. It is not a linear response, as it takes more output and SV fusion events to depolarize the muscle by 33 mV that would be accounted for by a simple addition of 10 and 23 mV like they suggest. This should be discussed.

6. The authors should clarify what has really already been done – both with respect to BoNT-C as well as in terms of selective Is and Ib manipulations. The authors do not appropriately cite the previous BoNT-C Backhaus et al., 2016 paper. This is misleading because Backhaus not only tested cleavage targets of BoNTs, but actually showed UAS-BoNT-C to disrupt transmission but also to kill the animals when expressed in muscle, suggesting non-SV fusion targets are also being affected.

---

## [Author Response]

Recommendations for Authors (Integrated review)1. Determine precisely when BoNT-C blocks synaptic transmission. Is embryonic transmission completely blocked or does the blockage occur later in development? This is critical for interpretation of the dataset.

We agree this is an important point and one that deserves further characterization. BoNT-C is a potent enzyme that cleaves synaptic SNARE complexes (Dong et al., 2019), so we expect that as soon as BoNT-C is synthesized it should work to rapidly and persistently block transmission. Therefore, it is important to establish when in development the three GAL4 drivers we use in this study (OK319, Is, and Ib) are expressed and confirm that transmission is blocked when this expression happens. Two complementary approaches discussed below indicate that BoNT-C silences transmission at early developmental stages (by at least 24H AEL).

First, we have driven a UAS-CD4::tdGFP reporter with each driver and determined when GFP is observed at NMJ terminals. In both OK319>GFP and Is>GFP, we observed strong neural expression by stage 17 embryos (data not shown), along with clear GFP staining at NMJ terminals by early first-instar larvae (24-28 hours after egg lay; AEL), which persisted through second instar stages (48H AEL). Similarly, with Ib>GFP, we observed weak but apparent neural GFP staining in stage 17 embryos and NMJ terminals at early first- and second-instar larvae. This data is shown in a new Supplementary Figure S2 and discussed.

Second, to assess whether synaptic transmission was silenced at early developmental stages, we have performed ca^2+^ imaging in postsynaptic compartments using the SynapGCaMP8f reporter we generated in this study. Although robust spontaneous activity was observed at wild-type NMJs at early first instar stages (24-28H AEL), no GCaMP8f signals were observed at Is or Ib NMJs of OK319>BoNT-C, Is NMJs of Is>BoNT-C, or Ib NMJs of Ib>BoNT-C. This data is shown in a new Supplementary Figure S2 and discussed.

2. The authors should do a more careful developmental study of synapse maturation, particularly PSD development, to assay if the block in synaptic transmission (assuming it occurs early on) leads to delay in PSD maturation, or simply has no effect. I'm not sure the authors can make their conclusion so strongly given they only assayed the NMJ in late 3rd instars. Assaying PSD length and GluRIIA/B distribution in 1st and 2nd instars, where they would have more temporal control to examine activity-dependent PSD development, would be useful. Perhaps at earlier time points the authors would see a delay in PSD/GluR field maturation, but that this difference is less apparent at older synapses where delays might be masked. If so, this would indicate activity can control the rate of GluR field assembly, but is not required for the process to eventually complete.

We agree that a more thorough analysis of PSD maturation at earlier 1^st^ and 2^nd^ instar stages would benefit this study. As previously reported, DLG structure at the NMJ is not well organized until 2^nd^ instar stages (Lahey et al., 1994; Mathew et al., 2002), so we focused our characterization on glutamate receptor maturation at the PSD. We find that GluRIIA and GluRIID levels and localization are similar to wild type controls following BoNT-C silencing at both first- and second-instar NMJs, including the mean fluorescence intensity, bouton number, GluR puncta size and number, and PSD area. These results are now presented in a new Supplementary Figure S6. While we find no obvious evidence to suggest that PSD maturation is delayed in BoNT-C silenced NMJs compared to wild type, we agree with the reviewers that we cannot rule out more subtle changes in the timing of synaptic development. Thus, we have also revised our language in relevant parts of the manuscript.

3. Please discuss why the block in synaptic transmission does not trigger the PHP response the authors have previously described with enhanced BRP accumulation at presynaptic AZs.

This is an astute point raised the reviewers and certainly one worthy of discussion. Genetic loss or pharmacological inhibition of postsynaptic glutamate receptors at the fly NMJ leads to the induction of retrograde signaling and enhanced presynaptic neurotransmitter release, referred to as Presynaptic Homeostatic Potentiation (PHP). As the reviewers point out, part of this homeostatic signaling system involves remodeling of the presynaptic active zone with an apparent increase in the size and intensity of active zone components, including BRP. Although the induction mechanism for how this process works is enigmatic, what is clear is that genetic loss or pharmacological perturbation of GluRIIA-containing receptors is absolutely required (Davis and Müller, 2015; Frank et al., 2020; Goel and Dickman, 2021). There is no evidence that loss or inhibition of presynaptic activity alone is capable of inducing PHP (or active zone remodeling), and in fact there is evidence that PHP can be induced in the absence of synaptic activity (Frank et al., 2006; Goel et al., 2017; Kikuma et al., 2019). Therefore, the active zone remodeling observed in PHP is not expected to be triggered in BoNT-C silenced synapses, since GluRIIA-containing receptors are present and unperturbed (Figure 4). We have added a discussion of these important points and thank the reviewers for raising this interesting issue.

4. Can the authors provide a discussion of why their BoNT-C does not block all post-Golgi traffic given Syx1 is required for both SV fusion and post-Golgi vesicle fusion with the plasma membrane.

This is also an astute and important point, and we agree the target of BoNT-C in neurons should be discussed more thoroughly. Although the reviewers are correct that Syx1 is involved in both regulated (synaptic vesicle fusion) and constitutive (post-Golgi) membrane trafficking in *Drosophila* (Schulze et al., 1995), there are 10 *Drosophila syntaxin* genes (and 3 Syx1A isoforms), which may be partially redundant in constitutive membrane trafficking (e.g. syx16; see (Xu et al., 2002)). Further, there is evidence that BoNT activity is variable depending on the tissue it is expressed (Dong et al., 2019), and cleavage assays in *Drosophila* S2 cells determined that each toxin cleaves its SNARE substrate with variable efficiency (Backhaus et al., 2016). Finally, clostridial SNARE toxins are known to preferentially target specific SNARE components when in particular complexes. For example, a new BoNT variant termed “BoNT-X” was recently shown to effectively proteolyze VAMP4/VAMP5/Ykt6. VAMP4 is found on Golgiderived vesicles that traffic to late endosomes, where it forms a trans-SNARE complex with the QSNARE complex Stx6/Stx7/Vti1b (Zhang et al., 2017). It seems that this processivity is unique for the BoNT-X/VAMP4 pairing and based on subtle differences in the protease recognition site of the substrate, rendering VAMP4 refractory to other BoNT serotypes. Hence, a similar BoNT-C/substrate mismatch may prevent cleavage of non-synaptic Syx components, at least in neurons. We have added a discussion of these important points.

5. The idea that 'composite values of Is and Ib neurotransmission can be fully recapitulated by isolated physiology from each input' (Figure 7) is not a new experiment and the idea of "reconstitution of wild-type NMJ physiology" by simply summing up evoked responses using sharp electrode recordings is questionable. Because they see a sum 33 mV EPSP and a IB of 9.5 mV and a Is of 23 mV doesn't really mean anything other than the Is has a stronger output, as has been previously shown by numerous labs. Without voltage clamp, it becomes harder and harder to depolarize the muscle as you get further from the resting potential. It is not a linear response, as it takes more output and SV fusion events to depolarize the muscle by 33 mV that would be accounted for by a simple addition of 10 and 23 mV like they suggest. This should be discussed.

Respectfully, we do not think it is accurate to assert that the component values of Is and Ib fully recapitulating wild type physiology (blended Is+Ib) shown in Figure 7 is “not a new experiment”. The reviewer is correct that qualitative, relative differences between Is vs Ib physiology have been reported previously (Hoang and Chiba, 2001; Karunanithi et al., 2002; Lnenicka and Keshishian, 2000; Lu et al., 2016; Pawlu et al., 2004), or evoked but not miniature transmission was separated (Genç and Davis, 2019; Sauvola et al., 2021). However, the term “neurotransmission” refers to quantitative, electrophysiological analysis of both miniature and evoked synaptic activity. To our knowledge, complete neurotransmission (fully isolated electrophysiological miniature and evoked activity) has never been achieved previously from entire MN-Is vs MN-Ib NMJs, nor shown to recapitulate the behavior of wild type (blended Is+Ib transmission). Hence, while previous studies have, for example, indeed shown that evoked transmission from MN-Is is stronger than from MN-Ib (Genç and Davis, 2019; Lu et al., 2016), miniature electrophysiological activity in these studies was not isolated, making it impossible to accurately determine quantal content and other fundamental parameters of synaptic function. Therefore, the data presented in Figure 7 is not only a new experiment which, for the first time, reveals complete, quantitatively accurate electrophysiological isolation of all neurotransmission from MN-Is or MN-Ib, but is perhaps the most important data to clearly establish baseline transmission from Is or Ib. Furthermore, this crucial data is *absolutely necessary* for any conclusions to be drawn about whether heterosynaptic functional plasticity is induced after manipulating Is or Ib. Indeed, the failure to accurately establish baseline synaptic function (miniature and evoked events) from Is vs Ib was a major confound in previous studies on heterosynaptic plasticity that precluded an accurate conclusion about whether heterosynaptic functional plasticity was actually observed.

The reviewer does bring up an important point related to nonlinear summation, which may obscure the actual depolarization induced when recorded in sharp electrode current clamp configuration. Generally, there are little voltage-gated currents induced when *Drosophila* larval muscles are depolarized less than ~40 mV (Miyamoto, 1978; Stevens, 1976), minimizing the potential impacts of non-linear summation, which is why the field continues to perform current clamp electrophysiology at the *Drosophila* NMJ at lowered extracellular ca^2+^ conditions (0.4 mM ca^2+^ in this study). However, we agree that two-electrode voltage clamp (TEVC) configuration would be a more accurate way to demonstrate the crucial point that composite values of Is and Ib neurotransmission can accurately match wild type (blended Is+Ib) transmission as shown in Figure 7.

Therefore, we have now included complete TEVC electrophysiology (miniature and evoked activity) in the same conditions as previously shown for current clamp data. This data demonstrates the same quantitative reconstitution of wild-type neurotransmission from composite Is and Ib isolated by BoNT-C silencing. This data is now presented in a new Supplementary Figure S7.

6. The authors should clarify what has really already been done – both with respect to BoNT-C as well as in terms of selective Is and Ib manipulations. The authors do not appropriately cite the previous BoNT-C Backhaus et al., 2016 paper. This is misleading because Backhaus not only tested cleavage targets of BoNTs, but actually showed UAS-BoNT-C to disrupt transmission but also to kill the animals when expressed in muscle, suggesting non-SV fusion targets are also being affected.

We appreciate these points, some of which was raised and responded to in point 4 above. First, we note that we have an open and collegial relationship with Prof Langenham (author on the Backhaus et al., paper) and have discussed several of these issues at length with him over the course of this study. Given our discussions, we would like to summarize several important points. First, all the transgenic BoNT transgenic fly lines described in Backhaus 2016 were lost by the time we contacted Prof Langenham in 2019 to discuss our preliminary findings on the BoNT transgenes we engineered and characterized in *Drosophila*. We had, ideally, wanted to compare the lines described in the 2016 study with our own, but were unable due to their loss. Second, Backhaus et al., found that each BoNT cleaves its substrate in S2 cells with variable efficiency, which is likely to be even more variable given the complex cellular environment in neurons and across diverse other tissues with a variety of SNARE complexes involved in regulated or constitutive trafficking. Third, only lethality and behavior (peristaltic movements and visiondependent motion) were used by Backhaus et al., to assess BoNT activity in vivo; *synaptic physiology was not*, *in fact*, *ever directly assessed*. Given these issues, it is difficult to compare the Backhaus et al., findings with the behavior of our BoNT-C in motor neurons in this study. That being said, we do agree that it is worthwhile to relay the following points and update the relevant areas in the manuscript.

First, like Backhaus et al., we observed variable phenotypes with the various BoNT transgenic flies we engineered, not only between individual BoNT transgenes but even comparing different inserts of the same transgene. For example, different BoNT-A and BoNT-E transgenic inserts were lethal when crossed to elav-Gal4 (pan-neuronal driver), while other inserts were viable. Furthermore, there was a rather large disconnect between the severity of lethality following expression of the transgene (in either neurons or muscle) and the impact on neurotransmission. For example, BoNT-B appeared to be highly toxic, requiring injection of >800 embryos to get a single viable insert. Although this BoNT-B insert was indeed highly toxic, leading to lethality when expressed in neurons, expression in motor neurons hardly impacted NMJ transmission (see Table S1). In contrast, we obtained a single BoNT-C transgene after injecting ~100 embryos, which was also lethal when expressed in all neurons. However, BoNT-C expression in motor neurons completely blocked all neurotransmission.

In terms of toxicity when expressed in non-neural tissue, we also observe that our BoNT-C transgene and other BoNT variants are lethal when expressed in muscle (Table S1). As discussed in our response to point 4 above, there are many potential SNARE targets for BoNT-C cleavage in muscle that may not be shared in motor neurons. Thus, the actual behavior of any specific BoNT transgene likely is impacted not only by the BoNT toxin itself but also influenced by other factors such as the strength of expression, transgenic insertion site, tissue in which it is expressed, specific SNARE complexes in each tissue, etc.

We have now more fully discussed these points in the revised methods and discussion.

References

Aponte-Santiago, N.A., Ormerod, K.G., Akbergenova, Y., and Littleton, J.T. (2020). Synaptic Plasticity Induced by Differential Manipulation of Tonic and Phasic Motoneurons in *Drosophila*. Journal of Neuroscience *40*, 6270–6288. https://doi.org/10.1523/JNEUROSCI.0925-20.2020.

Backhaus, P., Langenhan, T., and Neuser, K. (2016). Effects of transgenic expression of botulinum toxins in *Drosophila*. Journal of Neurogenetics *30*, 22–31. https://doi.org/10.3109/01677063.2016.1166223.

Chen, X., Ma, W., Zhang, S., Paluch, J., Guo, W., and Dickman, D.K. (2017). The BLOC-1 Subunit Pallidin Facilitates Activity-Dependent Synaptic Vesicle Recycling. ENeuro *4*. https://doi.org/10.1523/ENEURO.0335-16.2017.

Davis, G.W., and Müller, M. (2015). Homeostatic control of presynaptic neurotransmitter release. Annual Review of Physiology *77*, 251–270. https://doi.org/10.1146/annurev-physiol-021014-071740.

Delgado, R., Maureira, C., Oliva, C., Kidokoro, Y., and Labarca, P. (2000). Size of Vesicle Pools, Rates of Mobilization, and Recycling at Neuromuscular Synapses of a *Drosophila* mutant, shibire. Neuron *28*, 941–953. https://doi.org/10.1016/S0896-6273(00)00165-3.

Dickman, D.K., Horne, J.A., Meinertzhagen, I.A., and Schwarz, T.L. (2005). A Slowed Classical Pathway Rather Than Kiss-and-Run Mediates Endocytosis at Synapses Lacking Synaptojanin and Endophilin. Cell *123*, 521–533. https://doi.org/10.1016/J.CELL.2005.09.026.

Dong, M., Masuyer, G., and Stenmark, P. (2019). Botulinum and Tetanus Neurotoxins.

Https://Doi.Org/10.1146/Annurev-Biochem-013118-111654 *88*, 811–837. https://doi.org/10.1146/ANNUREV-BIOCHEM-013118-111654.

Frank, C.A., Kennedy, M.J., Goold, C.P.P., Marek, K.W., and Davis, G.W.W. (2006). Mechanisms Underlying the Rapid Induction and Sustained Expression of Synaptic Homeostasis. Neuron *52*, 663– 677. https://doi.org/10.1016/j.neuron.2006.09.029.

Frank, C.A., James, T.D., and Müller, M. (2020). Homeostatic control of *Drosophila* neuromuscular junction function. Synapse *74*, 22133. https://doi.org/10.1002/syn.22133.

Genç, Ö., and Davis, G.W. (2019). Target-wide Induction and Synapse Type-Specific Robustness of Presynaptic Homeostasis. Curr Biol *29*, 3863-3873.e2. https://doi.org/10.1016/J.CUB.2019.09.036.

Goel, P., and Dickman, D. (2021). Synaptic homeostats: latent plasticity revealed at the *Drosophila* neuromuscular junction. Cellular and Molecular Life Sciences *78*, 3159–3179. https://doi.org/10.1007/s00018-020-03732-3.

Goel, P., Li, X., and Dickman, D. (2017). Disparate Postsynaptic Induction Mechanisms Ultimately Converge to Drive the Retrograde Enhancement of Presynaptic Efficacy. Cell Reports *21*, 2339–2347. https://doi.org/10.1016/j.celrep.2017.10.116.

Heerssen, H., Fetter, R.D., and Davis, G.W. (2008). Clathrin Dependence of Synaptic-Vesicle Formation at the *Drosophila* Neuromuscular Junction. Current Biology *18*, 401–409. https://doi.org/10.1016/J.CUB.2008.02.055.

Hoang, B., and Chiba, A. (2001). Single-Cell Analysis of *Drosophila* Larval Neuromuscular Synapses.

Developmental Biology *229*, 55–70. https://doi.org/10.1006/DBIO.2000.9983.

Karunanithi, S., Marin, L., Wong, K., and Atwood, H.L. (2002). Quantal Size and Variation Determined by Vesicle Size in Normal and Mutant *Drosophila* Glutamatergic Synapses. Journal of Neuroscience *22*, 10267–10276. https://doi.org/10.1523/JNEUROSCI.22-23-10267.2002.

Kikuma, K., Li, X., Perry, S., Li, Q., Goel, P., Chen, C., Kim, D., Stavropoulos, N., and Dickman, D. (2019). Cul3 and insomniac are required for rapid ubiquitination of postsynaptic targets and retrograde homeostatic signaling. Nature Communications *10*, 1–13. https://doi.org/10.1038/s41467-019-10992-6.

Kuromi, H., and Kidokoro, Y. (1998). Two Distinct Pools of Synaptic Vesicles in Single Presynaptic Boutons in a Temperature-Sensitive *Drosophila* Mutant, shibire. Neuron *20*, 917–925. https://doi.org/10.1016/S0896-6273(00)80473-0.

Lahey, T., Gorczyca, M., Jia, X.X., and Budnik, V. (1994). The *Drosophila* tumor suppressor gene dlg is required for normal synaptic bouton structure. Neuron *13*, 823–835. https://doi.org/10.1016/08966273(94)90249-6.

Li, X., Goel, P., Wondolowski, J., Paluch, J., and Dickman, D. (2018). A Glutamate Homeostat Controls the Presynaptic Inhibition of Neurotransmitter Release. Cell Rep *23*, 1716–1727. https://doi.org/10.1016/J.CELREP.2018.03.130.

Littleton, J.T., Chapman, E.R., Kreber, R., Garment, M.B., Carlson, S.D., and Ganetzky, B. (1998).

Temperature-Sensitive Paralytic Mutations Demonstrate that Synaptic Exocytosis Requires SNARE Complex Assembly and Disassembly. Neuron *21*, 401–413. https://doi.org/10.1016/S08966273(00)80549-8.

Lnenicka, G.A., and Keshishian, H. (2000). Identified Motor Terminals in *Drosophila* Larvae Show Distinct Differences in Morphology and Physiology. J Neurobiol *43*, 186–197. https://doi.org/10.1002/(SICI)10974695(200005)43:2.

Lu, Z., Chouhan, A.K., Borycz, J.A., Lu, Z., Rossano, A.J., Brain, K.L., Zhou, Y., Meinertzhagen, I.A., and Macleod, G.T. (2016). High-Probability Neurotransmitter Release Sites Represent an Energy-Efficient Design. Current Biology *26*, 2562–2571. https://doi.org/10.1016/J.CUB.2016.07.032.

Mathew, D., Gramates, L.S., Packard, M., Thomas, U., Bilder, D., Perrimon, N., Gorczyca, M., and

Budnik, V. (2002). Recruitment of Scribble to the Synaptic Scaffolding Complex Requires GUK-holder, a Novel DLG Binding Protein. Current Biology *12*, 531–539. https://doi.org/10.1016/S0960-9822(02)00758-

3.

Miyamoto, M.D. (1978). Estimates on magnitude of nonlinear summation of evoked potentials at motor end plate. Https://Doi.Org/10.1152/Jn.1978.41.3.589 *41*, 589–599. https://doi.org/10.1152/JN.1978.41.3.589.

Pawlu, C., DiAntonio, A., and Heckmann, M. (2004). Postfusional Control of Quantal Current Shape. Neuron *42*, 607–618. https://doi.org/10.1016/S0896-6273(04)00269-7.

Sauvola, C.W., Akbergenova, Y., Cunningham, K.L., Aponte-Santiago, N.A., and Troy Littleton, J. (2021). The decoy snare tomosyn sets tonic versus phasic release properties and is required for homeostatic synaptic plasticity. *eLife 10*. https://doi.org/10.7554/eLife.72841.

Schulze, K.L., Broadie, K., Perin, M.S., and Bellen, H.J. (1995). Genetic and electrophysiological studies of *Drosophila* syntaxin-1A demonstrate its role in nonneuronal secretion and neurotransmission. Cell *80*, 311–320. https://doi.org/10.1016/0092-8674(95)90414-X.

Stevens, C.F. (1976). A comment on Martin’s relation. Biophysical Journal *16*, 891–895. https://doi.org/10.1016/S0006-3495(76)85739-6.

Verstreken, P., Kjaerulff, O., Lloyd, T.E., Atkinson, R., Zhou, Y., Meinertzhagen, I.A., and Bellen, H.J.

(2002). Endophilin mutations block clathrin-mediated endocytosis but not neurotransmitter release. Cell *109*, 101–112. https://doi.org/10.1016/S0092-8674(02)00688-8/ATTACHMENT/773E6B2A-9EDA-4C83B4F3-3514960BEA4C/MMC3.JPG.

Wang, Y., Lobb-Rabe, M., Ashley, J., Anand, V., and Carrillo, R.A. (2021). Structural and Functional Synaptic Plasticity Induced by Convergent Synapse Loss in the *Drosophila* Neuromuscular Circuit. Journal of Neuroscience *41*, 1401–1417. https://doi.org/10.1523/JNEUROSCI.1492-20.2020.

White, B., Osterwalder, T., and Keshishian, H. (2001). Molecular genetic approaches to the targeted suppression of neuronal activity. Current Biology *11*, R1041–R1053. https://doi.org/10.1016/S09609822(01)00621-2.

Xu, H., Boulianne, G.L., and Trimble, W.S. (2002). *Drosophila* syntaxin 16 is a Q-SNARE implicated in Golgi dynamics. Journal of Cell Science *115*, 4447–4455. https://doi.org/10.1242/JCS.00139.

Zhang, S., and Roman, G. (2013). Presynaptic Inhibition of Γ Lobe Neurons Is Required for Olfactory Learning in *Drosophila*. Current Biology *23*, 2519–2527. https://doi.org/10.1016/J.CUB.2013.10.043.

Zhang, S., Masuyer, G., Zhang, J., Shen, Y., Henriksson, L., Miyashita, S.I., Martínez-Carranza, M., Dong, M., and Stenmark, P. (2017). Identification and characterization of a novel botulinum neurotoxin. Nature Communications 2017 8:1 *8*, 1–10. https://doi.org/10.1038/ncomms14130.